

# Effects of variation in background mixing ratios of N₂, O₂, and Ar on the measurement of $\delta^{18}O$-H₂O and $\delta^2H$-H₂O values by cavity ring-down spectroscopy

Jennifer E. Johnson[1, 2] and Chris W. Rella[3]

[1] Department of Ecology and Evolutionary Biology, University of Arizona, Tucson, AZ, USA

[2] Current address: Department of Global Ecology, Carnegie Institution, Stanford, CA, USA

[3] Picarro, Inc., Santa Clara, CA, USA

10  *Correspondence to*: J. E. Johnson (jjohnson@carnegiescience.edu)

**Abstract.** Cavity ring-down spectrometers have generally been designed to operate under conditions in which the background gas has a constant composition. However, there are a number of observational and experimental situations of interest in which the background gas has a variable composition. In this study, we examine the effect of background gas composition on a cavity ring-down spectrometer that measures $\delta^{18}O$-H₂O and $\delta^2H$-H₂O values based

15  on the amplitude of water isotopologue absorption features around 7184 cm⁻¹ (L2120-*i*, Picarro, Inc.). For background mixtures balanced with N₂, the apparent $\delta^{18}O$-H₂O values deviate from true values by -0.50 ± 0.001 ‰ O₂ %⁻¹ and -0.57 ± 0.001 ‰ Ar %⁻¹, and apparent $\delta^2H$-H₂O values deviate from true values by 0.26 ± 0.004 ‰ O₂ %⁻¹ and 0.42 ± 0.004 ‰ Ar %⁻¹. The artifacts are the result of broadening, narrowing, and shifting of both the target absorption lines and strong neighboring lines. While the background-induced isotopic artifacts can largely be

20  corrected with simple empirical or semi-mechanistic models, neither type of model is capable of completely correcting the isotopic artifacts to within the inherent instrument precision. The development of strategies for dynamically detecting and accommodating background variation in N₂, O₂, and/or Ar would facilitate the application of cavity ring-down spectrometers to a new class of observations and experiments.



## 1 Introduction

In most commercially available laser absorption spectrometers, the accuracy and precision of trace gas measurements are sensitive to the composition of the background gas. In this paper, we explore this issue in the context of a class of laser absorption spectrometers that is of increasing importance for environmental research, the cavity ring-down spectroscopy (CRDS) analyzers. While the CRDS analyzers can accurately and precisely measure the concentration and isotopic composition of trace gases in situations where the background gas has a constant composition, they make substantial measurement errors in situations where the background gas has a variable composition (Aemisegger et al., 2012; Becker et al., 2012; Chen et al., 2010; Friedrichs et al., 2010; Long et al., 2013; Nara et al., 2012; Volkmann and Weiler, 2014). In variable backgrounds, measurement errors emerge from the interaction between two factors: first, collisional shifting and broadening of the trace gas absorption transitions; and second, the spectral acquisition and analysis strategies employed by the CRDS analyzers (Gralher et al., 2016; Hendry et al., 2011; Sprenger et al., 2017). While the fundamental collisional effects are qualitatively well-understood, their quantitative impacts on analyzer performance and the strategies needed to overcome those impacts are both incompletely understood.

To date, background effects on CRDS measurements have been reported in three different types of situations. First, calibrations for observations of the unconfined atmosphere: even though the natural levels of variability in atmospheric $N_2$, $O_2$, and Ar mixing ratios are small, significant contrasts can occur between the average composition of the atmosphere and the composition of the mixtures used for calibration (Aemisegger et al., 2012; Chen et al., 2010; Long et al., 2013; Nara et al., 2012). Second, observations of confined atmospheres: for trace gas measurements in lakes, streams, oceans, and soils, the background concentrations of $O_2$ can vary naturally over a wide range because the rates of biological processes that produce and consume this gas can proceed more rapidly than the physical processes that control mixing with the unconfined atmosphere (Becker et al., 2012; Friedrichs et al., 2010). Third, experiments with active control of background composition: some measurement techniques utilize $N_2$ dilution to modulate the concentrations of target trace gases in both confined and unconfined atmospheric backgrounds (Gralher et al., 2016; Volkmann and Weiler, 2014).

The fundamental physical mechanisms that give rise to background gas effects on CRDS measurements are well-understood. The CRDS analyzers use high-finesse optical cavities to make ultra-sensitive quantitative absorption measurements based on infrared absorption transitions of various trace gases (O'Keefe and Deacon, 1988). Two features of the absorption transitions of the trace gases are affected by collisions with the background gas: (i) the frequencies of maximum absorption intensity (i.e., denoted $\nu_0$), and (ii) the shapes of the absorption line profiles around those central frequencies (i.e., described by $I(\nu_0)$, the maximum amplitude at $\nu_0$, and $I(\nu_0)/2$, the full-width at half-maximum) (Demtroder, 2014; Hanson et al., 2015). The former effect is termed 'shifting'; the latter effect is termed 'broadening' and/or 'narrowing.' Due to these effects, any contrast between the backgrounds used for calibrations versus observations changes the geometry of the target absorption spectrum and has the potential to introduce errors into the resulting measurements of trace gas concentrations and isotope ratios. However, whether or not errors actually occur is a function of how specific CRDS analyzers are designed.





In principle, it should be possible to make CRDS measurements that are completely insensitive to background gas composition by measuring the integrated absorbance (i.e., the absorption peak area) of any isolated absorption feature in a given spectrum (Zalicki and Zare, 1995). In practice, however, most current-generation CRDS analyzers are expected to exhibit some degree of sensitivity to background gas composition because they: (i)

target absorption features that are not completely isolated from neighboring absorption features; (ii) measure the amplitude, rather than the area, of the target absorption features; and/or (iii) attempt to optimize measurement precision by treating lineshape parameters as fixed rather than free variables (Hendry et al., 2011; Hodges and Lisak, 2006; Steig et al., 2014). On account of these design constraints, the susceptibility of different CRDS analyzers to background gas effects is a function of the identity of the specific absorption features that are targeted, the spectral

acquisition approach that is used to measure those features, and the spectral analysis techniques that are used to interpret the measurements. The interactions between these factors make it difficult to predict how any particular analyzer will respond to background gas variation. As a result, experimental measurements are necessary to determine both the quantitative impacts of background gas variation on analyzer performance and the strategies needed to overcome those impacts.

The overall objective of this study is to characterize how background gas composition affects measurements of water isotopologues in one commercially-available CRDS analyzer, the L2120-$i$ manufactured by Picarro, Inc. Three factors make the L2120-$i$ an attractive test bed for studying background gas effects. First, a number of other types of interference have been studied in the L2120-$i$. Previous work has characterized interference from self-broadening (Schmidt et al., 2010) and from organic contaminants (Brand et al., 2009; West et al., 2010),

tested algorithms for correcting for organic interference during or after analysis (Hendry et al., 2011; Johnson et al., 2017; Martín-Gómez et al., 2015; Schmidt et al., 2012; Schultz et al., 2011; West et al., 2011), and tested peripherals for pyrolizing or oxidizing organic contaminants prior to analysis (Berkelhammer et al., 2013; Lazarus et al., 2016; Martín-Gómez et al., 2015). Second, the L2120-$i$ has been widely used to measure $\delta^{18}$O-$H_2O$ and $\delta^2$H-$H_2O$ values in situations where background variation could be relevant to the calibration procedures and/or the fundamental

measurements. Examples include applications to measurements of liquid water in precipitation (Munksgaard et al., 2011), plant water (West et al., 2011), soil water (Herbstritt et al., 2012), and seawater (Munksgaard et al., 2012), as well as water vapor in the terrestrial boundary layer (Berkelhammer et al., 2013) and marine boundary layer (Steen-Larsen et al., 2014). Third, it has recently been shown that the L2120-$i$ measurements are highly sensitive to the $N_2/O_2$, $N_2/CO_2$, and $CO_2/O_2$ composition of the background gas, and that the magnitude of the sensitivity is relevant

to many observational and experimental situations (Gralher et al., 2016).

To evaluate how background gas composition impacts L2120-$i$ measurements and the strategies needed to correct for those impacts, we carried out a series of experiments addressing the following questions:

(i)      What are the magnitudes of the effects of variation in the mixing ratio of $N_2/O_2$, $N_2/Ar$, and $O_2/Ar$

35          on the apparent $\delta^{18}$O-$H_2O$ and $\delta^2$H-$H_2O$ values measured by the L2120-$i$ CRDS analyzer?





(ii)     How are the background effects on apparent $\delta^{18}O$-$H_2O$ and $\delta^{2}H$-$H_2O$ values derived from the interaction between the target spectra and the spectral acquisition and analysis strategies in this instrument?

(iii)    Is it practicable to develop *post hoc* calibrations for this instrument that accurately account for the effects of background variation in $N_2$, $O_2$, and/or Ar on the apparent $\delta^{18}O$-$H_2O$ and $\delta^{2}H$-$H_2O$ values?

## 2 Methods

### 2.1 Background gas mixtures

Background gas streams with various compositions of $N_2$, $O_2$, and Ar were generated with a mixing system (Figure 1). The mixing system consisted of four cylinders of compressed gas, thermal mass flow controllers, and a backpressure regulator upstream of the CRDS instrument inlet. Three of the cylinders contained ultra high-purity $N_2$, $O_2$, and Ar (99.999 % purity, <3 ppm $H_2O$, and <0.5 ppm total hydrocarbon content (THC); ALPHAGAZ 1, Air Liquide America Specialty Gases LLC, Houston, TX, USA). The fourth cylinder contained ultra high-purity air (<1 ppm $H_2O$, <0.01 ppm THC, <0.01 ppm CO, <0.001 ppm $NO_x$, <0.001 ppm $SO_2$; Ultrapure Air, Scott-Marrin, Inc., Riverside, CA, USA) with the $N_2$, $O_2$, and Ar composition of the natural atmosphere (i.e., 78.1 % $N_2$, 20.9 % $O_2$, 0.9 % Ar; (Brewer et al., 2014; Flores et al., 2015)). In the experiments, background gas mixtures were dynamically mixed from these cylinders with the mass flow controllers (FC-260 with RO-28, Tylan-Mykrolis, Allen, TX, USA). The mass flow controllers were calibrated with a bubble flow meter (25 mL Kimax bubble flow tube, Kimble-Chase, Vineland, NJ, USA) and mixing accuracy was tested for $N_2/O_2$ and $Ar/O_2$ mixtures with a galvanic oxygen sensor (MO-200, Apogee Instruments, Logan, UT, USA). With this system, the composition of each mixture could be controlled to an accuracy of ± 0.1 % of each constituent. The back-pressure regulator was used to ensure that the mixtures were supplied to the CRDS analyzer inlet at 2.5 psi above atmospheric pressure.

### 2.2 Liquid water standards

All of the measurements in this study were based on four vaporized liquid standards. The isotopic composition of the standards was initially established by measurement with a Finnigan Delta S Isotope Ratio Mass Spectrometer (Thermo Fisher Scientific, West Palm Beach, FL, USA) in the Environmental Isotope Laboratory, Department of Geosciences, University of Arizona (Tucson, AZ, USA). For oxygen, samples were equilibrated with $CO_2$ gas at approximately 15° C in an automated equilibration device coupled to the mass spectrometer. For hydrogen, samples were reacted at 750° C with Cr metal using a Finnigan H/Device coupled to the mass spectrometer. Standardization was based on distilled water standards referenced to VSMOW2 and SLAP2.

The resulting standards had the following isotopic compositions: (1) $\delta^{18}O$ = -3.74 ‰, $\delta^{2}H$ = -15.3 ‰; (2) $\delta^{18}O$ = -9.52 ‰, $\delta^{2}H$ = -62.2 ‰, (3) $\delta^{18}O$ = -14.18 ‰, $\delta^{2}H$ = -102.7 ‰, (4) $\delta^{18}O$ = -30.32 ‰, $\delta^{2}H$ = -246.7 ‰. These values were determined with analytical precision of ± 0.08 ‰ for $\delta^{18}O$-$H_2O$ and ± 0.9 ‰ for $\delta^{2}H$-$H_2O$. To ensure that the isotopic composition of the standards remained stable over time, they were stored in 1L amber glass bottles,

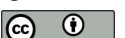



with Polyseal cone-lined screw caps, sealed with Parafilm. For CRDS measurements, 1.5 mL aliquots of each standard were pipetted into 1.8 mL glass vials with polypropylene screw caps and bonded PTFE-silicone septa (66020-950 and 46610-700, VWR, Radnor, PA, USA). To eliminate any effects from diffusive losses through the septa, each vial was measured within 24 hours of being filled and was sampled for a maximum of n = 10 successive

injections.

### 2.3 Spectral acquisition

The CRDS analyzer used in these experiments was an L2120-*i* (Picarro, Inc., Santa Clara, CA, USA). The key components of this analyzer are a laser, a wavelength monitor, an optical cavity, and a photodetector. The laser targets $H_2O$ absorption lines close to 7184 $cm^{-1}$ (1392 nm). The specific lines that are utilized are 7183.685 $cm^{-1}$

(1392.043 nm) for $^1H^1H^{16}O$, 7183.585 $cm^{-1}$ (1392.063 nm) for $^1H^1H^{18}O$, and 7183.972 $cm^{-1}$ (1391.988 nm) for $^1H^2H^{16}O$ (Tennyson et al., 2009; 2010; 2013). Operationally, the analyzer scans the laser across these features, recording absorption loss as a function of optical frequency (spectrograms). To generate each frequency and absorption pair, light from the laser is directed into the optical cavity, the frequency is determined by the wavelength monitor, and the power in the cavity is monitored with a photodetector detecting light leaking through one of the

mirrors. The absorption is quantified based on the rate at which the light intensity decays (i.e., 'rings down') when the laser is turned off (Crosson, 2008).

Since the absorbance measurements are of gas-phase $H_2O$, a front-end peripheral must be used to convert liquid-phase standards into the gas-phase (Gupta et al., 2009). For this study, the L2120-*i* analyzer was equipped with a V1102-i high-precision vaporizer (Picarro, Inc., Santa Clara, CA, USA) and autosampler (HTC PAL, Leap

Technologies, Carrboro, NC, USA). To ensure stable performance, the analyzer, vaporizer, and autosampler were installed in an air-conditioned laboratory where the air temperature was maintained at 20.0 ± 1.7° C. All of the measurements were performed in the air carrier mode, and with the vaporizer running at 110°C. Injections were made on a 9 minute cycle, using a 10 uL syringe (SGE 10R-C/T-5/0.47C, Trajan Scientific Americas, Inc., Austin, TX, USA) which was rinsed twice in N-methyl-2-pyrrolidone (99.5 %, Acros Organics, Fisher Scientific,

Pittsburgh, PA, USA) before each injection.

### 2.4 Spectral analysis

In the L2120-*i*, the spectrograms are interpreted with a non-linear curve fitting routine based on the Levenberg-Marquardt algorithm. The analysis is conceptually similar to that utilized in the earlier generation L1102-*i* analyzers (Hendry et al., 2011), but some details differ. Briefly, the fitting routine compares each measured

spectrogram to a modeled spectrogram and adjusts the model parameters in order to minimize the residual error. The modeled spectrogram represents a mixture of $^1H^1H^{16}O$, $^1H^1H^{18}O$, and $^1H^2H^{16}O$ in a pure water standard (i.e., one that was evaluated in a zero air background and had an isotopic composition near $\delta^{18}O = 0$ ‰ and $\delta^2H = 0$ ‰). The fitting routine compares this modeled spectrogram to the measured spectrogram in three stages.

In the first stage, the fitting routine varies the amounts of $^1H^1H^{16}O$, $^1H^1H^{18}O$, and $^1H^2H^{16}O$, the centration

and scale of the frequency axis, the absolute value and slope of the baseline, the linewidth, and the amounts of



several potential organic contaminants (CH$_4$, C$_2$H$_6$, and MeOH). This fit determines the observed centration and scale of the frequency axis ('h2o_shift' and 'h2o_squish_a'), the observed linewidth ('h2o_y_eff_a'), and the linewidth expected for the observed amount of $^1$H$^1$H$^{16}$O in an air background ('h2o_y_eff'). The second and third stages of fitting then make different assumptions about the presence of organic contaminants.

5       In the second stage, the fitting routine assumes that there is no organic contamination. The centration and scale of the frequency axis are fixed based on the results of the first stage ('h2o_shift' and 'h2o_squish_a') and the effective linewidth is fixed based on the amount of $^1$H$^1$H$^{16}$O ('h2o_y_eff'). The free parameters are the amounts of $^1$H$^1$H$^{16}$O, $^1$H$^1$H$^{18}$O, and $^1$H$^2$H$^{16}$O, as well as the absolute value and slope of the baseline. This fit determines the reported residuals ('standard_residuals'), baseline ('standard_base'), baseline slope ('standard_slope'), H$_2$O mixing
ratio (from the amplitude of the $^1$H$^1$H$^{16}$O peak) and the $\delta^{18}$O and $\delta^2$H values (from the ratios of the amplitudes of the $^1$H$^1$H$^{16}$O, $^1$H$^1$H$^{18}$O, and $^1$H$^2$H$^{16}$O peaks).

      In the third stage, the fitting routine allows for the possibility of organic contamination. Here, the centration and scale of the frequency axis are fixed based on the results of the first stage ('h2o_shift' and 'h2o_squish_a') and the effective linewidth is fixed based on the observed linewidth ('h2o_y_eff_a'). The free parameters are the
amounts of the organic contaminants, the amounts of $^1$H$^1$H$^{16}$O, $^1$H$^1$H$^{18}$O, and $^1$H$^2$H$^{16}$O, as well as the absolute value and slope of the baseline. This fit determines the reported 'organic-corrected' residuals ('organic_res'), baseline ('organic_base'), baseline slope ('organic_slope'), and $\delta^{18}$O and $\delta^2$H values (from the ratios of the organic-corrected amplitudes of the three peaks).

## 2.5 Experimental design

### 2.5.1 Characterizing background gas effects

      We performed three experiments to characterize the effects of variation in the mixing ratios of N$_2$/O$_2$, N$_2$/Ar, and O$_2$/Ar, respectively. In each experiment, we generated five backgrounds from the two gases (i.e., 0/100, 25,75, 50/50, 75/25, and 100/0; in %). In each background, we measured the four liquid standards across a range of injection volumes (i.e., 400-2400 nL, in eleven steps of 200 nL each). For the most isotopically enriched standard,
we performed three replicate injections at each injection volume; for the other three standards, we performed a single injection at each injection volume. At each transition between standards, we inserted fifteen additional injections to allow for full equilibration and eliminate any carryover effects. At each transition between backgrounds, we inserted four additional injections in UHP N$_2$ to check for instrumental drift. Both the transition injections and the drift check injections were analyzed solely for quality control and quality assurance purposes. For
the primary analyses of N$_2$, O$_2$, and Ar effects, the remaining injections yielded a total sample size of n = 330 per experiment, and n = 990 across the three experiments.

### 2.5.2 Evaluating corrections for background gas effects

      We performed a fourth experiment to test whether the background effects observed in the pure gases (N$_2$, O$_2$, Ar) and binary gas mixtures (N$_2$/O$_2$, N$_2$/Ar, and O$_2$/Ar) could be used to predict the effects in a ternary mixture



representing natural atmospheric composition ($N_2/O_2/Ar$). In this experiment, we used the ultra high-purity whole air as the background and measured the same four liquid standards across a range of injection volumes (i.e., 600, 1200, 2000, 3000 nL). For each standard, we performed three replicate injections at each injection volume. At each transition between standards, we again inserted fifteen additional injections to allow for full equilibration and

eliminate any carryover effects. The transition injections were analyzed solely for quality control and quality assurance purposes, such that the remaining injections yielded a total sample size of n = 240.

**2.6 Data analysis**

Since the default configuration of the L2120-*i* software does not write all of the intermediate spectral parameters to the liquid injection output files, we retrieved the analyzer's complete raw data files for the duration of

these experiments from the archive directory, and calculated the mean and standard deviation of each parameter over the intervals defined by the injection peak-picking algorithm. Statistical analyses were then performed using the open-source statistical software, R (R Core Team, 2016). Briefly, the data were fit to a series of multivariate linear models using the 'lm()' function from base R, and the fit was evaluated in terms of the residual standard error (RSE), adjusted $R^2$ and F-test P-value. More details of each analysis are provided in the following sections.

**2.6.1 Notation**

For all samples, the relative abundances of the heavy and light isotopologues were expressed with the dimensionless isotope ratios:

$$R_{sample} = [^1H^1H^{18}O]/[^1H^1H^{16}O], \text{ or } [^1H^2H^{16}O]/[^1H^1H^{16}O], \tag{1}$$

The isotope ratios were normalized relative to the international standard VSMOW (Vienna Standard Mean Ocean Water):

$$\delta^{18}O_{apparent} \text{ or } \delta^2H_{apparent} \text{ (‰)} = (R_{sample}/R_{VSMOW} - 1), \tag{2}$$

where $R_{sample}$ and $R_{VSMOW}$ represent the ratios of the abundance of the heavy and light isotopologues in the samples and international standard, respectively.

**2.6.2 Magnitude of background effects**

To visualize the effects of variation in the mixing ratios of $N_2/O_2$, $N_2/Ar$, and $O_2/Ar$ on the apparent

isotopic composition of $H_2O$, we plotted the background composition against the difference between the apparent and true isotopic composition of each sample (i.e., $\Delta\delta^{18}O$-$H_2O = \delta^{18}O$-$H_2O_{apparent} - \delta^{18}O$-$H_2O_{true}$ and $\Delta\delta^2H$-$H_2O = \delta^2H$-$H_2O_{apparent} - \delta^2H$-$H_2O_{true}$). To quantify the magnitude of the effects of background variation on the apparent isotopic composition of $H_2O$, we then fit a series of multivariate linear models. First, we examined the





measurements in the pure background gases. For each pure background, we described variation in the apparent isotopic composition of water with a multivariate linear model of the form:

$$Y = \beta_0 + X_1 \cdot \beta_1 + X_2 \cdot \beta_2 + X_3 \cdot \beta_3 + X_4 \cdot \beta_4, \qquad (3)$$

where Y is the apparent isotopic composition of water ($\delta^{18}O\text{-}H_2O_{apparent}$, $\delta^2H\text{-}H_2O_{apparent}$; in units ‰), $X_1$ is the true isotopic composition of water ($\delta^{18}O\text{-}H_2O_{true}$, $\delta^2H\text{-}H_2O_{true}$; in units ‰), $X_2$ is the water mixing ratio ($H_2O$; in units %), $X_3$ is the inverse of the water mixing ratio ($1/X_2$), $X_4$ is the square of the water mixing ratio ($X_2^2$), and $\beta_0$, $\beta_1$, $\beta_2$, $\beta_3$, and $\beta_4$ are the regression coefficients. Next, we examined the measurements in the binary mixtures. For each binary

mixture, we added an additional term to capture the effects of background variation:

$$Y = \beta_0 + X_1 \cdot \beta_1 + X_2 \cdot \beta_2 + X_3 \cdot \beta_3 + X_4 \cdot \beta_4 + X_5 \cdot \beta_5, \qquad (4)$$

where $X_5$ is the mixing ratio of either $O_2$ or $Ar$ (in units %) and $\beta_5$ is the corresponding regression coefficient.

Finally, we combined all of the binary mixtures into a composite dataset. For this composite dataset, we added two terms to capture the effects of background variation:

$$Y = \beta_0 + X_1 \cdot \beta_1 + X_2 \cdot \beta_2 + X_3 \cdot \beta_3 + X_4 \cdot \beta_4 + X_5 \cdot \beta_5 + X_6 \cdot \beta_6, \qquad (5)$$

where $X_5$ and $X_6$ are the mixing ratios of $O_2$ and $Ar$ (in units %) and $\beta_5$ and $\beta_6$ are the corresponding regression coefficients.

### 2.6.4 Geometric basis of background effects

To visualize the geometric basis of the background effects, we plotted the $\Delta\delta^{18}O\text{-}H_2O$ and $\Delta\delta^2H\text{-}H_2O$ values against three parameters calculated during the second stage of fitting ('standard_residuals', 'standard_base',
'standard_slope'), three parameters calculated during the first stage of fitting and included as fixed values during the second stage ('h2o_shift', 'h2o_squish_a', 'h2o_y_eff'), and one parameter calculated during the first stage of fitting and omitted during the second stage ('h2o_y_eff_a'). To visualize the interactions between the background composition and the $H_2O$ mixing ratio, we also plotted each parameter against the $H_2O$ mixing ratio ('h2o_ppmv'). We then formulated a series of semi-mechanistic models to test which of the seven spectral parameters was the best
predictor of the isotopic error terms. Each model described variation in the apparent isotopic composition of water as:

$$Y = \beta_0 + X_1 \cdot \beta_1 + X_2 \cdot \beta_2 + X_3 \cdot \beta_3 + X_4 \cdot \beta_4 + X_7 \cdot \beta_7, \qquad (6)$$



where $X_7$ is one of the seven spectral parameters and $\beta_7$ is the corresponding regression coefficient. N.B., this expression is analogous to Eq. 5 with the exception that one of the spectral parameters ($X_7$) has been substituted for the mixing ratios of $O_2$ and Ar ($X_5$ and $X_6$).

### 2.6.5 Prediction of background effects in a ternary mixture

We used the empirical model described by Eq. (5) and the semi-mechanistic model described by Eq. (6) derived from the measurements of the standards in binary mixtures to predict the apparent isotopic composition of the standards within the ternary gas mixture. To evaluate how water vapor self-broadening vs. background-broadening affected model performance, we assessed model skill across the entire range of water vapor mixing ratios (*i.e.*, n = 240 analyses for 2,500 ppmv ≤ $H_2O$ ≤ 35,000 ppmv), as well as within a restricted subset of

intermediate-range water vapor mixing ratios (*i.e.*, n = 116 analyses for 10,000 ppmv ≤ $H_2O$ ≤ 25,000 ppmv). To provide a benchmark for evaluating the $1\sigma$ precision of each empirical and semi-mechanistic model, we calculated the long-term $1\sigma$ precision of the L2120-*i* analyzer using an independent dataset comprised of previous measurements of the same set of standards, across the same range of water mixing ratios, and in the same type of ultra-high purity air that was used to test the two models.

## 3 Results

### 3.1 Magnitude of background effects

Across the $N_2$/$O_2$, $N_2$/Ar, and $O_2$/Ar mixing experiments, the average differences between the apparent and true isotopic composition of the standards are -18.45 ± 20.02 ‰ for $\Delta\delta^{18}$O-$H_2O$ and 24.5 ± 17.1 for $\Delta\delta^2$H-$H_2O$ (i.e.,

for n = 990; Figure 2). The variation in $\Delta\delta^{18}$O-$H_2O$ and $\Delta\delta^2$H-$H_2O$ is partially due to the variation in the mixing ratio of $H_2O$, and partially due to the variation in the mixing ratios of $N_2$/$O_2$, $N_2$/Ar, and $O_2$/Ar. For $\delta^{18}$O-$H_2O$, the range of $H_2O$ mixing ratios that was evaluated has effects of smaller magnitude than the ranges of $N_2$/$O_2$, $N_2$/Ar, and $O_2$/Ar mixing ratios that were evaluated (Figure 2, a-c). For $\delta^2$H-$H_2O$, both factors have effects of similar magnitude (Figure 2, d-f).

Within the subsets of measurements made in pure $N_2$, $O_2$, and Ar, Eq. (3) accounts for the effects of the $H_2O$ mixing ratio with overall precision ranging between 0.37-1.16 ‰ for $\delta^{18}$O-$H_2O$ and 2.3-3.7 for $\delta^2$H-$H_2O$ (Table 1). The structure of the best-fit models varies between isotopologues and between backgrounds, with all three of the $H_2O$ mixing ratio-dependent parameters significant in some cases and none significant in others (Table 1). Within the models where the coefficient describing the first-order response to the $H_2O$ mixing ratio, $\beta_2$, has significant

explanatory power, it tends to have a negative sign for $\delta^{18}$O-$H_2O$ (Table 1; Models 1, 3) and a positive sign for $\delta^2$H-$H_2O$ (Table 1; Model 4).

Within each of the binary mixtures, Eq. (4) accounts for the combined effects of the $H_2O$ mixing ratio and the $N_2$/$O_2$, $N_2$/Ar, and $O_2$/Ar mixing ratios with overall precision ranging between 0.33-0.62 ‰ for $\delta^{18}$O-$H_2O$ and 3.2-5.2 for $\delta^2$H-$H_2O$ (Table 2). Within these models, the coefficient describing the first-order response to the $O_2$ or

Ar mixing ratio, $\beta_5$, also tends to have a negative sign for $\delta^{18}$O-$H_2O$ (Table 2; Models 1, 3, 5) and a positive sign for





$\delta^2$H-H$_2$O (Table 2; Models 2, 4, 6). For both $\delta^{18}$O-H$_2$O and $\delta^2$H-H$_2$O, the magnitude of $\beta_5$ in the O$_2$/Ar experiment is equivalent to the difference in the magnitude of $\beta_5$ in the N$_2$/O$_2$ experiment versus in the N$_2$/Ar experiment (Table 2).

When all three experiments are combined into a single dataset, Eq. (5) accounts for the combined effects of the H$_2$O mixing ratio and the N$_2$/O$_2$, N$_2$/Ar, and O$_2$/Ar mixing ratios with overall precision of 0.62 ‰ for $\delta^{18}$O-H$_2$O and 3.6 ‰ for $\delta^2$H-H$_2$O (Table 3). Within these models, the coefficients describing the first-order response to the O$_2$ and Ar mixing ratios, $\beta_5$ and $\beta_6$, have negative signs for $\delta^{18}$O-H$_2$O (Table 3; Model 1) and positive signs for $\delta^2$H-H$_2$O (Table 3; Model 2). For both $\delta^{18}$O-H$_2$O and $\delta^2$H-H$_2$O, the sensitivity to Ar is relatively higher than the sensitivity to O$_2$ (Table 3; Models 1, 2). In absolute terms, the apparent $\delta^{18}$O-H$_2$O values deviate from true values by -0.50 ± 0.001 ‰ O$_2$ %$^{-1}$ and -0.57 ± 0.001 ‰ Ar %$^{-1}$, respectively (Table 3; Model 1). The apparent $\delta^2$H-H$_2$O values deviate from true values by 0.26 ± 0.004 ‰ O$_2$ %$^{-1}$ and 0.42 ± 0.004 ‰ Ar %$^{-1}$, respectively (Table 3; Model 2).

### 3.2 Geometric basis of background effects

Overall, the relationships between the isotopic error terms and the spectral parameters have two shared features. First, the background composition has consistent effects across the three experiments. For each spectral fitting parameter, the patterns observed in the O$_2$/Ar experiment are equivalent to the difference in the patterns observed in the N$_2$/O$_2$ experiment versus in the N$_2$/Ar experiment (i.e., for $\delta^{18}$O values, compare differences between panels (a) and (b) to (c) in Figures 3-9; for $\delta^2$H values, compare differences between panels (d) and (e) to (f) in Figures 3-9). Second, for those spectral parameters that have significant linear relationships with the isotopic error terms, the relative sensitivities of the two isotopologues to any given spectral parameter always have opposing signs. The apparent $\delta^{18}$O values become more depleted with decreases in the absolute value of the baseline (Figure 4, a-c), increases in the slope of the baseline (Figure 5, a-c), increases in the frequency scale correction parameter (Figure 7, a-c), and decreases in the free linewidth parameter (Figure 9, a-c). In contrast, the apparent $\delta^2$H values become more enriched with the analogous changes in those parameters (Figures 3-9, d-f).

Beyond these shared features, there is substantial variation between the spectral parameters in terms of the complexity of their relationships to the isotopic error terms. The spectral residuals do not have a linear relationship with the isotopic error terms: although maximum values of the residuals are usually associated with maximum values of the isotopic error terms, minimum values of the residuals are associated with the full range of values of the isotopic error terms (Figure 3, a-f). The absolute value of the baseline and the baseline slope each have significant linear relationships with the isotopic error terms, but also have large amounts of variation that are not directly related to the isotopic error terms (Figures 4-5, a-f). The frequency shift parameter does not have significant relationships with the isotopic error terms (Figure 6, a-f), but the frequency scale correction parameter does have significant linear relationships with the isotopic error terms (Figure 7, a-f). Analogously, the fixed linewidth parameter does not have significant relationships with the isotopic error terms (Figure 8, a-f), but the free linewidth parameter does have significant relationships with the isotopic error terms (Figure 9, a-f).

The complexity of the relationships between the spectral parameters and isotopic error terms is driven by interactions between the background composition and the H$_2$O mixing ratios (Figures 10-11). For the spectral



residuals, baseline, and baseline slope, there is a multiplicative interaction between the background composition and $H_2O$ mixing ratio: at the lowest $H_2O$ mixing ratios, variation in background composition has the smallest effects on the spectral residuals, baseline, and baseline slope; at the highest $H_2O$ mixing ratios, the opposite is true (Figure 10, a-c). For the frequency scale correction and free linewidth parameters, there is an additive interaction between the

background composition and $H_2O$ mixing ratio: regardless of the $H_2O$ mixing ratio, variation in background composition has similar effects on the frequency scale correction and free linewidth parameters (Figure 11, b, d). For the frequency shift and fixed linewidth parameters, there is no interaction between the background composition and $H_2O$ mixing ratio: both parameters vary with the $H_2O$ mixing ratio, but those relationships are insensitive to the background composition (Figure 11, a, c).

In the semi-mechanistic models (Eq. 6), the free linewidth parameter is a better predictor than any of the other spectral parameters (Table 4). The models based on the free linewidth parameter account for the combined effects of the $H_2O$ mixing ratio and the $N_2/O_2$, $N_2/Ar$, and $O_2/Ar$ mixing ratios with overall precision of 2.54 ‰ for $\delta^{18}O$-$H_2O$ and 4.4 ‰ for $\delta^2H$-$H_2O$ (Table 4). In these models, the coefficient describing the first-order response to the free linewidth parameter, $\beta_7$, has a positive sign for $\delta^{18}O$-$H_2O$ (Table 4; Model 7) and a negative sign for $\delta^2H$-

$H_2O$ (Table 4; Model 14). The second-best predictor is the frequency scale correction parameter (Table 4). The models based on the frequency scale correction parameter have overall precision of 6.72 ‰ for $\delta^{18}O$-$H_2O$ and 4.9 ‰ for $\delta^2H$-$H_2O$ (Table 4). For this parameter, the $\beta_7$ coefficient has a negative sign for $\delta^{18}O$-$H_2O$ (Table 4; Model 6) and a positive sign for $\delta^2H$-$H_2O$ (Table 4; Model 13).

### 3.3 Prediction of background effects in a ternary mixture

For $\delta^{18}O$-$H_2O$, the empirical model predicts the apparent $\delta^{18}O$ values with a $1\sigma$ precision of ± 0.99 ‰, whereas the semi-mechanistic model predicts the apparent $\delta^{18}O$ values with a $1\sigma$ precision of ± 1.68 ‰ (n = 240; Table 5). When the test measurements in the ternary gas mixture are restricted to intermediate mixing ratios in the range of 10,000-25,000 ppmv $H_2O$, these values improve to ± 0.80 ‰ and ± 1.21 ‰, respectively (n = 116; Table 5). For $\delta^2H$-$H_2O$, the empirical model predicts the apparent $\delta^2H$ values with a $1\sigma$ precision of ± 3.1 ‰, whereas the

semi-mechanistic model predicts the apparent $\delta^2H$ values with a $1\sigma$ precision of ± 3.0 ‰ (Table 5). For intermediate mixing ratios in the range of 10,000-25,000 ppmv $H_2O$, these values improve to ± 2.0 ‰ and ± 2.1 ‰, respectively (n = 116; Table 5). As a benchmark for comparison, the average long-term $1\sigma$ precision of this L2120-$i$ analyzer is ± 0.24 ‰ for $\delta^{18}O$-$H_2O$ and ± 1.4 ‰ for $\delta^2H$-$H_2O$ across the range of mixing ratios from 2,500 - 35,000 ppmv $H_2O$.

**4 Discussion**

**4.1 What are the magnitudes of the effects of variation in the mixing ratio of $N_2/O_2$, $N_2/Ar$, and $O_2/Ar$ on the apparent $\delta^{18}O$-$H_2O$ and $\delta^2H$-$H_2O$ values measured by the L2120-$i$ CRDS analyzer?**

Across the range of backgrounds considered in this study, variation in the $N_2/O_2$, $N_2/Ar$, and $O_2/Ar$ ratios has substantial effects on the apparent isotopic composition of water reported by the L2120-$i$ (Figure 2). For $\delta^{18}O$-

$H_2O$, combining the long-term $1\sigma$ precision of the analyzer (± 0.24 ‰) and the magnitude of the sensitivities to $O_2$





and Ar relative to $N_2$ (i.e., -0.50 ± 0.001 ‰ $O_2$ %$^{-1}$ and -0.57 ± 0.001 ‰ Ar %$^{-1}$; Table 3) implies that variation over the thresholds of ± 0.48 % $O_2$ or ± 0.42 % Ar is expected to result in detectable oxygen isotope errors. For $\delta^2$H-$H_2$O, combining the long-term 1σ precision of the analyzer (± 1.4 ‰) and the magnitude of the sensitivities to $O_2$ and Ar relative to $N_2$ (i.e., 0.26 ± 0.004 ‰ $O_2$ %$^{-1}$ and 0.42 ± 0.004 ‰ Ar %$^{-1}$; Table 3) implies that variation over the thresholds of ± 5.4 % $O_2$ or ± 3.3 % Ar is expected to result in detectable hydrogen isotope errors.

The only previous measurements available for direct comparison to these results are those of Gralher et al. (2016). In binary $N_2/O_2$ mixtures, Gralher et al. (2016) found that a different L2120-$i$ analyzer exhibited a sensitivity of -0.56 ‰ $O_2$ %$^{-1}$ for $\delta^{18}$O-$H_2$O and a sensitivity of 0.42 ‰ $O_2$ %$^{-1}$ for $\delta^2$H-$H_2$O (i.e., see Figure 2 in that reference). Overall, the Gralher et al. (2016) values are more similar to our binary $N_2$/Ar sensitivities (i.e., for $\delta^{18}$O-$H_2$O, -0.56 ± 0.001 ‰ Ar %$^{-1}$; for $\delta^2$H-$H_2$O, 0.43 ± 0.005 ‰ Ar %$^{-1}$; Table 2) than our binary $N_2/O_2$ sensitivities (i.e., for $\delta^{18}$O-$H_2$O, -0.50 ± 0.001 ‰ $O_2$ %$^{-1}$; for $\delta^2$H-$H_2$O, 0.26 ± 0.009 ‰ $O_2$ %$^{-1}$; Table 2). This is unexpected, and the responsible mechanisms are not entirely clear.

Since the Gralher et al. (2016) sensitivities were derived from measurements across narrow ranges of $H_2$O mixing ratios (i.e., ~ 17,000 ppmv) and $O_2$ mixing ratios (i.e., 0-20%), one possible explanation is that the wider ranges used in our study could be responsible for the different sensitivity estimates. Subsetting our dataset to the range of $H_2$O mixing ratios used by Gralher et al. (2016) yields $N_2/O_2$ sensitivities equivalent to those reported in Table 2. However, subsetting our dataset to the range of $O_2$ mixing ratios used by Gralher et al. (2016) does yield $N_2/O_2$ sensitivities in much closer agreement with those authors' results (i.e., for $\delta^{18}$O-$H_2$O, -0.54 ± 0.003 ‰ $O_2$ %$^{-1}$; for $\delta^2$H-$H_2$O, 0.38 ± 0.002 ‰ $O_2$ %$^{-1}$). This suggests that the differences in the sensitivity estimates are derived from the different ranges of $N_2/O_2$ mixing ratios used in the two studies.

### 4.2 How are the background effects on the apparent $\delta^{18}$O-$H_2$O and $\delta^2$H-$H_2$O values derived from the interaction between the target spectra and the spectral acquisition and analysis strategies in this instrument?

Overall, the strongest direct effect of the background gas composition is on the effective linewidth of the target absorption features. In pure $N_2$ backgrounds, the actual value of the effective linewidth is greater than the value prescribed for air backgrounds. The spectral analysis algorithm attempts to compensate for this mis-specification of the peak shape by decreasing the frequency scale correction parameter (i.e., h2o_squish_a), but the peak shape mis-specification persists in spite of the frequency scale adjustments (i.e., h2o_y_eff_a > h2o_y_eff). As a result, the amplitudes of the absorption peaks are systematically overestimated (Figure 12, a). The degree of overestimation increases in the order $^1H^2H^{16}O < ^1H^1H^{16}O < ^1H^1H^{18}O$, with the result that the $\Delta\delta^{18}$O-$H_2$O values are positive and the $\Delta\delta^2$H-$H_2$O values are negative. In pure $O_2$ and Ar backgrounds, the actual value of the effective linewidth is less than the value prescribed for air backgrounds. Here, the spectral analysis algorithm attempts to compensate by increasing the frequency scale correction parameter (i.e., h2o_squish_a), but the peak shape mis-specification again persists (i.e., h2o_y_eff_a < h2o_y_eff). In these backgrounds, the amplitudes of the absorption peaks are systematically underestimated (Figure 12, b). The degree of underestimation increases in the order $^1H^2H^{16}O < ^1H^1H^{16}O < ^1H^1H^{18}O$, such that the $\Delta\delta^{18}$O-$H_2$O values are negative and the $\Delta\delta^2$H-$H_2$O values are positive. On the one hand, the tendency for the absorption spectrum to be broader in $N_2$, intermediate in $O_2$, and narrower in





Ar is entirely consistent with the normal behavior of isolated water vapor absorption lines (Buldyreva et al., 2011). On the other hand, it is not entirely clear why the $^1H^2H^{16}O$, $^1H^1H^{16}O$, and $^1H^1H^{18}O$ lines exhibit increasing susceptibility to peak shape mis-specification.

One possible explanation is that the differential errors are derived from indirect effects of the background gas. The three water vapor absorption lines that are targeted by the L2120-$i$ are all characterized by relatively low line-strengths, but they sit on the upper 'wings' of lower-frequency water vapor absorption lines that are characterized by much higher line-stengths (i.e., at $\nu_0$ = 7181.156, 7182.209, and 7182.950 cm$^{-1}$; (Lisak et al., 2009); Figure 12, c). The broadening, narrowing, and shifting of these strong off-screen lines appear to be the major control on variation in the spectral residuals, spectral baseline, and baseline slope parameters. Specifically, N$_2$-induced increases in the width of the offscreen lines seem to decrease the baseline slope and increase the absolute value of the baseline, whereas O$_2$- and Ar-induced decreases in the width of the offscreen lines seem to increase the baseline slope and decrease the absolute value of the baseline. Since the target $^1H^1H^{18}O$ line is at a lower frequency than the target $^1H^1H^{16}O$ line, and the target $^1H^1H^{16}O$ line is in turn at a lower frequency than the target $^1H^2H^{16}O$ line, the baseline perturbations from the off-screen lines increase in the same rank order as, and could be responsible for, the differential peak shape mis-specifications (i.e., $^1H^2H^{16}O < {}^1H^1H^{16}O < {}^1H^1H^{18}O$).

However, proximity to the off-screen features is not the only possible explanation for the differential peak shape mis-specifications. For example, the $^1H^2H^{16}O$, $^1H^1H^{16}O$, and $^1H^1H^{18}O$ lines also vary in line strength in the order $^1H^2H^{16}O < {}^1H^1H^{16}O < {}^1H^1H^{18}O$ (Tennyson et al., 2009; 2010; 2013). As a result, a second possible explanation is that the differential susceptibility to peak shape mis-specification is primarily a function of line strength. This interpretation is supported by the fact that the largest negative $\Delta\delta^{18}O$-H$_2$O errors occur in the most $\delta^{18}O$-enriched standard (i.e., where the difference in amplitude between the $^1H^1H^{16}O$ and $^1H^1H^{18}O$ lines is maximized), whereas the largest positive $\Delta\delta^2H$-H$_2$O errors occurr in the most $\delta^2H$-depleted standard (i.e., where the difference in amplitude between $^1H^2H^{16}O$ and $^1H^1H^{16}O$ is maximized). Nonetheless, it could also be the case that the differential susceptibility to peak shape mis-specification is a function of a combination of several of the above mechanisms, or other undefined mechanisms. To definitively distinguish among these possibilities, it would be necessary to have accurate measurements of the broadening, narrowing, and shifting coefficients for each of the individual lines in the target spectrum, rather than the 'effective' coefficients that the L2120-$i$ calculates for the composite spectrum.

**4.3 Is it practicable to develop *post hoc* calibrations for this instrument that accurately account for the effects of background variation in N$_2$, O$_2$, and/or Ar on the apparent $\delta^{18}O$-H$_2$O and $\delta^2H$-H$_2$O values?**

On the one hand, the majority of the background-induced isotope artifacts can be corrected with either simple empirical or semi-mechanistic models (Table 5). The success of both types of models is likely a reflection of the fact that the collisional broadening, narrowing and shifting coefficients of any given absorption line in a mixed background can all be satisfactorily described as linear combinations of the corresponding coefficients in pure backgrounds (Buldyreva et al., 2011). On the other hand, neither type of model is capable of completely correcting the isotopic artifacts to within the inherent instrument precision (Table 5). Although the loss of precision for $\delta^2H$-H$_2$O is similar for the semi-mechanistic and empirical corrections, the loss of precision for $\delta^{18}O$-H$_2$O is slightly





greater for the semi-mechanistic corrections than for the empirical corrections. In combination, these findings indicate that there are several feasible approaches for *post hoc* calibrations of CRDS measurements that accurately account for background variation in $N_2$, $O_2$, and/or Ar, but that all currently tradeoff with measurement precision. This has important implications for the range of strategies that can be used to calibrate CRDS analyzers for observations (i) in the atmosphere, as well as (ii) in other settings.

For atmospheric applications, there are likely to be systematic inaccuracies in $\delta^{18}O$-$H_2O$ and *d*-excess values if 'synthetic air' is used as a calibration background without accounting for the fact that these $N_2/O_2$ mixtures lack Ar and may exhibit cylinder-to-cylinder variation in $N_2$ versus $O_2$ content. For example, these effects may explain the cylinder-to-cylinder calibration shifts observed when Air Liquide's 'ALPHAGAZ 1' has been used as a calibration background for atmospheric observations (e.g., see Aemisegger et al. (2012) and Casado et al. (2016)). To address this issue, previous studies have recommended performing CRDS calibrations for atmospheric observations in natural air backgrounds (Aemisegger et al., 2012; Chen et al., 2010; Long et al., 2013; Nara et al., 2012). The results of the current study corroborate this approach, but indicate that it represents only one of two alternatives. The other approach is performing calibration measurements in a background that does not conform to atmospheric composition, and using sensitivity experiments of the sort reported here to develop transfer functions that translate between the calibration and observation backgrounds (i.e., similar to Eq. 5). Despite its relatively lower precision, this approach may nonetheless represent the preferred strategy for applications where it is difficult or impossible to obtain sufficiently purified natural air for calibration.

For marine, freshwater, and soil applications, there are likely to be systematic inaccuracies in $\delta^{18}O$-$H_2O$, $\delta^2H$-$H_2O$, and *d*-excess values of liquid and vapor samples if the calibration strategy does not account for dynamic variation in the $O_2$ content of the measurement background. For example, marine dissolved oxygen levels range from supersaturated during high-productivity periods in upwelling zones (Schmidt and Eggert, 2016), to hypoxic during harmful algal blooms in coastal zones (O'Boyle et al., 2016), to anoxic in deep water oxygen minimum zones (Larsen et al., 2016). To address this type of dynamic variation in background $O_2$ content, Friedrichs et al. (2010) and Becker et al. (2012) have demonstrated that linewidth information from CRDS measurements of $CO_2$ and $\delta^{13}C$-$CO_2$ can be used to both detect and correct for $O_2$–induced errors. The results of the current study indicate that an analogous approach can be used with the L2120-*i* (i.e., based on Eq. 6), although doing so will further reduce the precision of the $\delta^{18}O$-$H_2O$ values. It is likely that the spectroscopically-based corrections are less successful in the L2120-*i* because the 7183-7184 cm$^{-1}$ region is congested, and the fitting algorithm does not perform individual fits on the target $H_2O$ isotopologue peaks. In contrast, the EnviroSense 2050 analyzer used by Friedrichs et al. (2010) and Becker et al. (2012) targeted a relatively uncongested spectral region (6251-6252 cm$^{-1}$) and performed individual fits on each of the target $CO_2$ isotopologue peaks.

Looking forward, the most straightforward approach to overcome the tradeoff between background stability and measurement precision would be to develop new spectral acquisition and analysis strategies for CRDS measurements that can accommodate dynamic variation in the composition of the background gas. Considering that the integrated absorbance of isolated features in CRDS spectra is expected to be conserved regardless of the degree of broadening, narrowing, or shifting induced by the background gas (Zalicki and Zare, 1995), the next generation of



CRDS analyzers that quantify absorption based on peak areas may be less sensitive to background variation than those that quantify absorption based on peak amplitudes (Steig et al., 2014). A recent report of the insensitivity of off-axis integrated cavity output spectroscopy (OA-ICOS) to background variation from 1-5 % $CO_2$ is consistent with this idea (Sprenger et al., 2017). However, while measurements of integrated absorbance may be sufficient for

limiting sensitivity to background effects, they are unlikely to be sufficient for entirely eliminating sensitivity to background effects. To develop this capability for environmental research, it may be helpful to introduce spectral fitting strategies that are similar to the 'calibration-free' spectral fitting strategies that have been recently been developed for high-temperature and high-pressure applications in energy research (Goldenstein et al., 2017; 2014; Sun et al., 2013; Sur et al., 2015).

**5 Conclusions**

      Due to the sensitivity of the L2120-*i* to background gas composition, this model CRDS analyzer is best suited for applications in which the background $O_2$ and Ar mixing ratios vary by no more than a maximum of 0.5 %, and ideally by less than 0.1 %. Calibration strategies should be designed to ensure that if there is any contrast between the background used for calibration and measurement, it is no greater than this threshold. For observations

or experiments in which the background $O_2$ and Ar mixing ratios vary by more than 0.5 %, the measurements of the L2120-*i* will include systematic errors that are derived from the broadening, narrowing, and shifting of both the target absorption lines and strong neighboring lines. If the composition of the variable background is known, the errors can be accurately corrected with the empirical model described by Eq. (5). If the composition of the variable background is unknown, the errors can also be accurately corrected with the semi-mechanistic model described by

Eq. (6). In either case, accuracy and precision will be maximized by calculating the coefficients for Eq. (5) or (6) from a calibration dataset that encompasses the full range of variation in $N_2$, $O_2$, and/or Ar mixing ratios, $H_2O$ mixing ratios, and $\delta^{18}O$-$H_2O$ and $\delta^2H$-$H_2O$ values within the unknown observations. Since neither of the *post hoc* correction approaches optimize precision, new strategies for dynamically detecting and accommodating background variation in $N_2$, $O_2$, and/or Ar are needed in order to capitalize on the possibilities of CRDS measurements in

variable backgrounds.

**Data availability**

      The data generated in this study are available through the Open Science Framework (DOI: 10.17605/OSF.IO/C7NSG).

**Author contributions**

30       J. E. Johnson and C. W. Rella designed the experiments and J. E. Johnson carried them out. Both authors analyzed the results and J. E. Johnson prepared the manuscript with contributions from C. W. Rella.



**Competing interests**

J. E. Johnson declares no conflict of interest. C. W. Rella is an employee of Picarro, Inc.

**Acknowledgements**

We are grateful to R. K. Monson (U. of Arizona) for encouragement to pursue these experiments; to J. A.
Berry (Carnegie Institution) and C. B. Field (Carnegie Institution) for loaning the Picarro L2120-*i*; to C. Redondo
(U. of Arizona), C. Burkhart (Air Liquide), S. Martinez (Air Liquide), and S. Owens (Air Liquide) for assistance
obtaining compressed gases; and to D. Dettman (U. of Arizona), F. Dominguez (U. of Illinois), C. Eastoe (U. of
Arizona), S. Leavitt (U. of Arizona), and K. Welten (U. of California-Berkeley) for assistance obtaining and
calibrating liquid standards. This study was completed with support from the National Science Foundation through
the Macrosystem Biology Program Award #1065790 (R.K.M.) and the Major Research Infrastructure Program
Award #1040106 (C.B.F. and J.A.B.).

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





**List of Figures**

**Figure 1.** Schematic diagram of the mixing system used for the experiments. Pure background gases were obtained commercially, mixed with mass flow controllers, and supplied to the inlet of the L2120-*i* at a slight

overpressure. See text for details.

**Figure 2.** Sensitivity of $\delta^{18}$O-$H_2O$ and $\delta^2$H-$H_2O$ values to background $N_2$, $O_2$, and Ar mixing ratios. For each experiment, sensitivity is plotted as the difference between the apparent and true isotopic composition of the standards (i.e., $\Delta\delta^{18}$O-$H_2O$ = $\delta^{18}$O-$H_2O_{apparent}$ - $\delta^{18}$O-$H_2O_{true}$ and $\Delta\delta^2$H-$H_2O$ = $\delta^2$H-$H_2O_{apparent}$ − $\delta^2$H-$H_2O_{true}$). For each panel, n = 330 measurements; points represent mean values ± s.d.

**Figure 3.** Relationship between the spectral residuals, $\Delta\delta^{18}$O-$H_2O$, and $\Delta\delta^2$H-$H_2O$. For each panel, n = 330 measurements; points represent mean values ± s.d.

**Figure 4.** Relationship between the spectral baseline, $\Delta\delta^{18}$O-$H_2O$, and $\Delta\delta^2$H-$H_2O$. For each panel, n = 330 measurements; points represent mean values ± s.d.

**Figure 5.** Relationship between the baseline slope, $\Delta\delta^{18}$O-$H_2O$, and $\Delta\delta^2$H-$H_2O$. For each panel, n = 330

measurements; points represent mean values ± s.d.

**Figure 6.** Relationship between the frequency shift, $\Delta\delta^{18}$O-$H_2O$, and $\Delta\delta^2$H-$H_2O$. For each panel, n = 330 measurements; points represent mean values ± s.d.

**Figure 7.** Relationship between the frequency scale correction, $\Delta\delta^{18}$O-$H_2O$, and $\Delta\delta^2$H-$H_2O$. For each panel, n = 330 measurements; points represent mean values ± s.d.

**Figure 8.** Relationship between the fixed linewidth parameter, $\Delta\delta^{18}$O-$H_2O$, and $\Delta\delta^2$H-$H_2O$. For each panel, n = 330 measurements; points represent mean values ± s.d.

**Figure 9.** Relationship between the free linewidth parameter, $\Delta\delta^{18}$O-$H_2O$, and $\Delta\delta^2$H-$H_2O$. For each panel, n = 330 measurements; points represent mean values ± s.d.

**Figure 10.** Relationships between $H_2O$ mixing ratio and the spectral residuals, baseline, and baseline slope

parameters. For each panel, n = 990 measurements; points represent mean values ± s.d.

**Figure 11.** Relationships between $H_2O$ mixing ratio and the frequency shift, frequency scale correction, fixed linewidth, and free linewidth parameters. For each panel, n = 990 measurements; points represent mean values ± s.d.



**Figure 12.** Direct and indirect effects of background gas composition on peak amplitude determination. Panels (a) and (b) plot theoretical spectrograms illustrating how isolated absorption features are directly broadened by air (solid line) versus $N_2$ (dashed line) or $O_2$ and Ar (dotted line). Panel (c) plots simulated spectrograms illustrating how the baseline of the three $H_2O$ lines targeted by the L2120-*i* is indirectly affected by the strong neighboring lines at lower frequencies. Simulations were performed using spectraplot.com with HITRAN/HITEMP data and the following parameters: 0.5 and 2.5 % $H_2O$, T = 80°C, P = 35 torr, and L = 1 cm.





**Figure 1.** Schematic diagram of the mixing system used for the experiments. Pure background gases were obtained commercially, mixed with mass flow controllers, and supplied to the inlet of the L2120-*i* at a slight overpressure. See text for details.

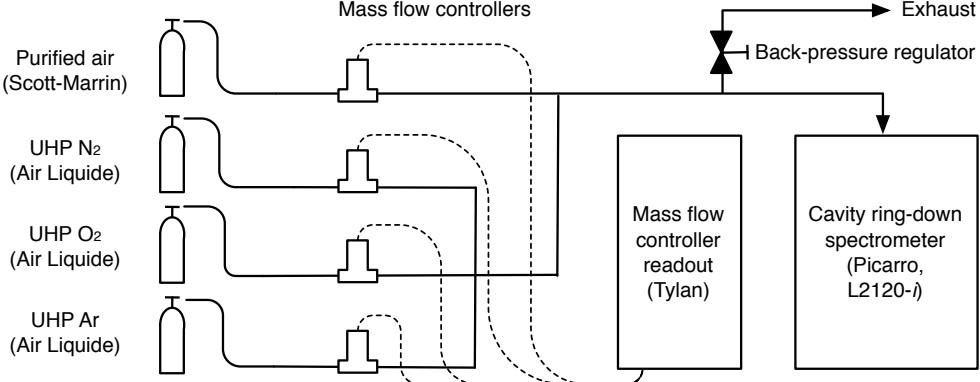





**Figure 2.** Sensitivity of $\delta^{18}O$-$H_2O$ and $\delta^2H$-$H_2O$ values to background $N_2$, $O_2$, and Ar mixing ratios. For each experiment, isotopic errors are plotted as the difference between the apparent and true isotopic composition of the standards (i.e., $\Delta\delta^{18}O$-$H_2O = \delta^{18}O$-$H_2O_{apparent} - \delta^{18}O$-$H_2O_{true}$ and $\Delta\delta^2H$-$H_2O = \delta^2H$-$H_2O_{apparent} - \delta^2H$-$H_2O_{true}$). For each panel, n = 330 measurements; points represent mean values ± s.d.; and regression slopes are given by $\beta_5$ values in Table 2.

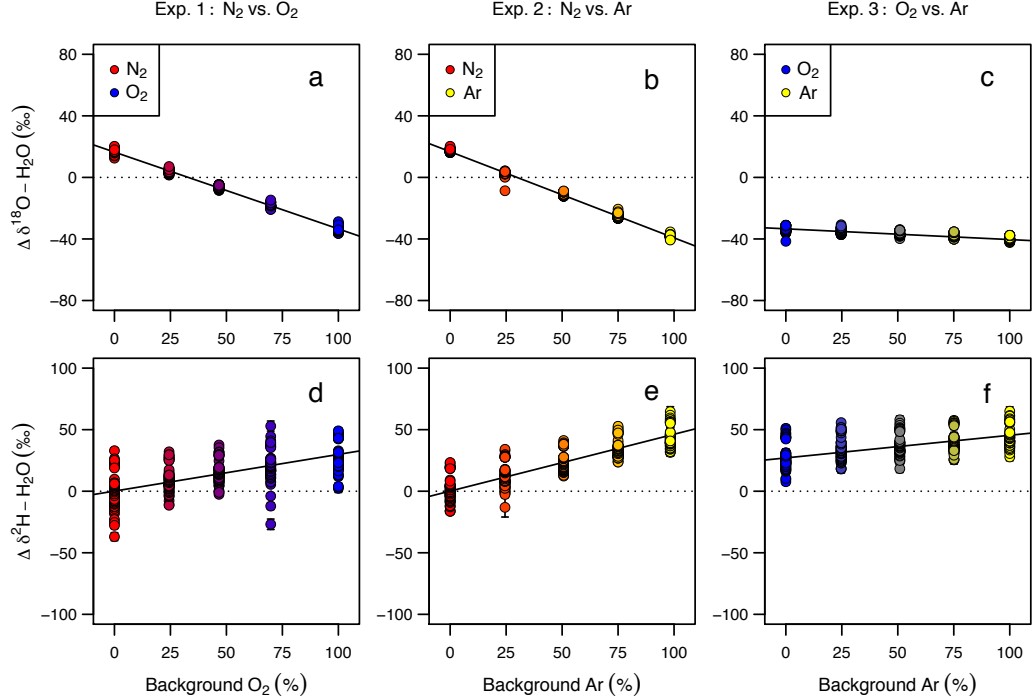



**Figure 3.** Relationship between the spectral residuals, $\Delta\delta^{18}$O-$H_2O$, and $\Delta\delta^{2}$H-$H_2O$. For each panel, n = 330 measurements; points represent mean values ± s.d.

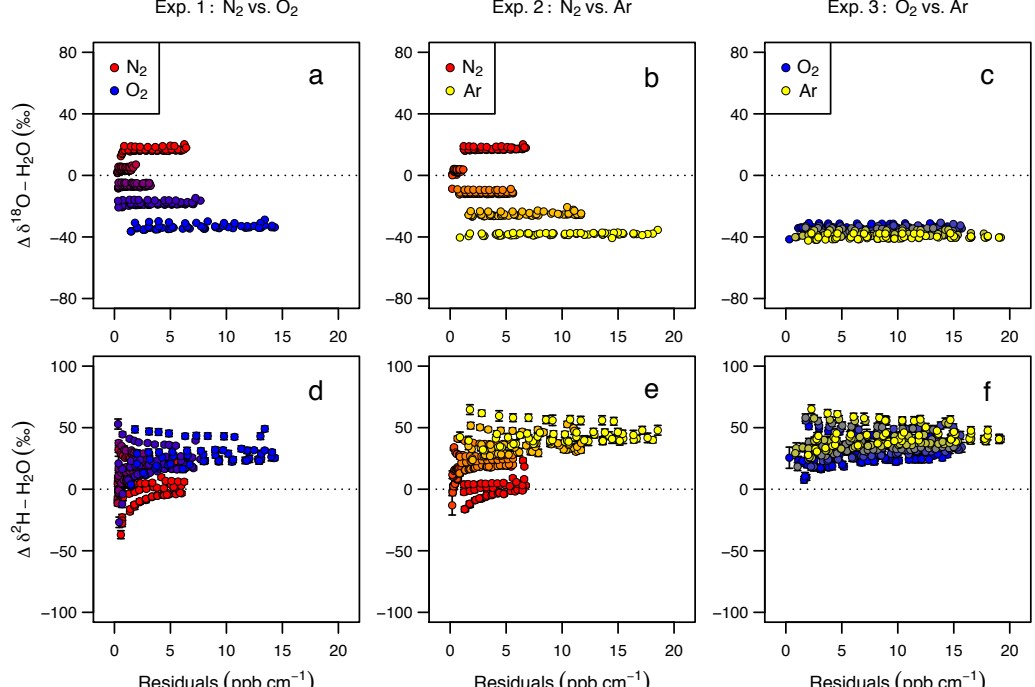



**Figure 4.** Relationship between the spectral baseline, $\Delta\delta^{18}O$-$H_2O$, and $\Delta\delta^2H$-$H_2O$. For each panel, n = 330 measurements; points represent mean values ± s.d.

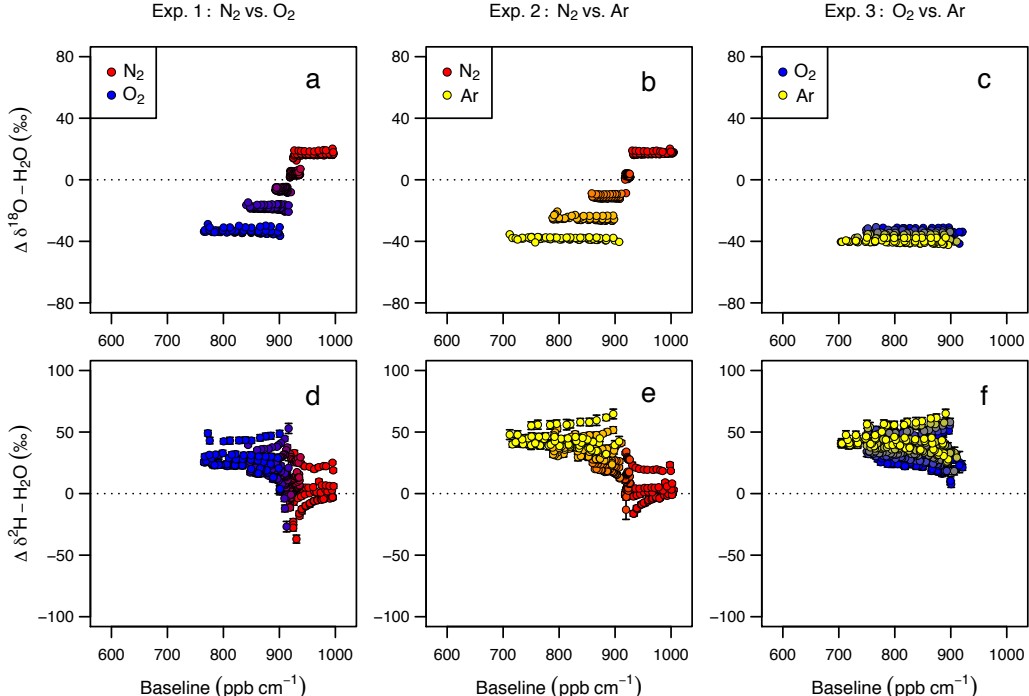



**Figure 5.** Relationship between the baseline slope, $\Delta\delta^{18}$O-$H_2O$, and $\Delta\delta^{2}$H-$H_2O$. For each panel, n = 330 measurements; points represent mean values ± s.d.

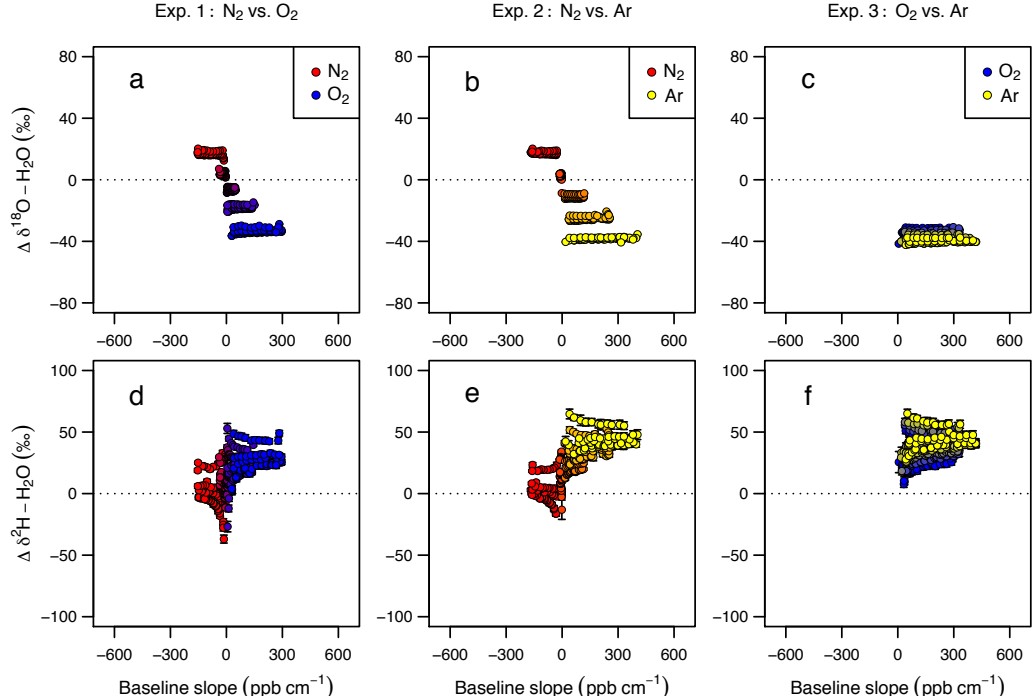





**Figure 6.** Relationship between the frequency shift, $\Delta\delta^{18}$O-$H_2O$, and $\Delta\delta^{2}$H-$H_2O$. For each panel, n = 330 measurements; points represent mean values ± s.d.

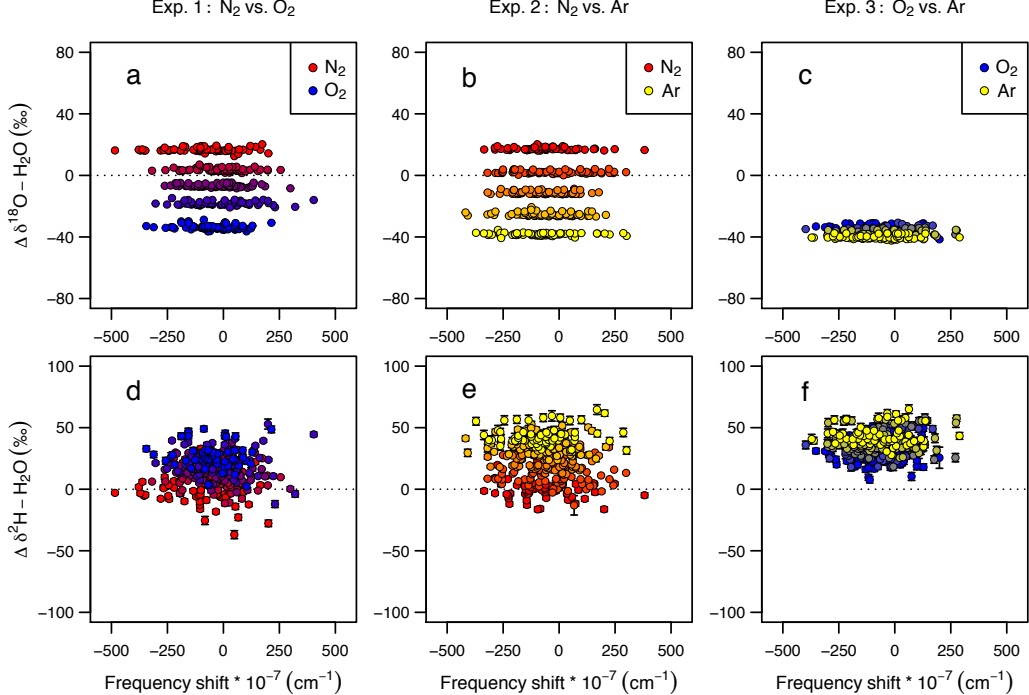



**Figure 7.** Relationship between the frequency scale correction, $\Delta\delta^{18}O$-$H_2O$, and $\Delta\delta^2H$-$H_2O$. For each panel, n = 330 measurements; points represent mean values ± s.d.

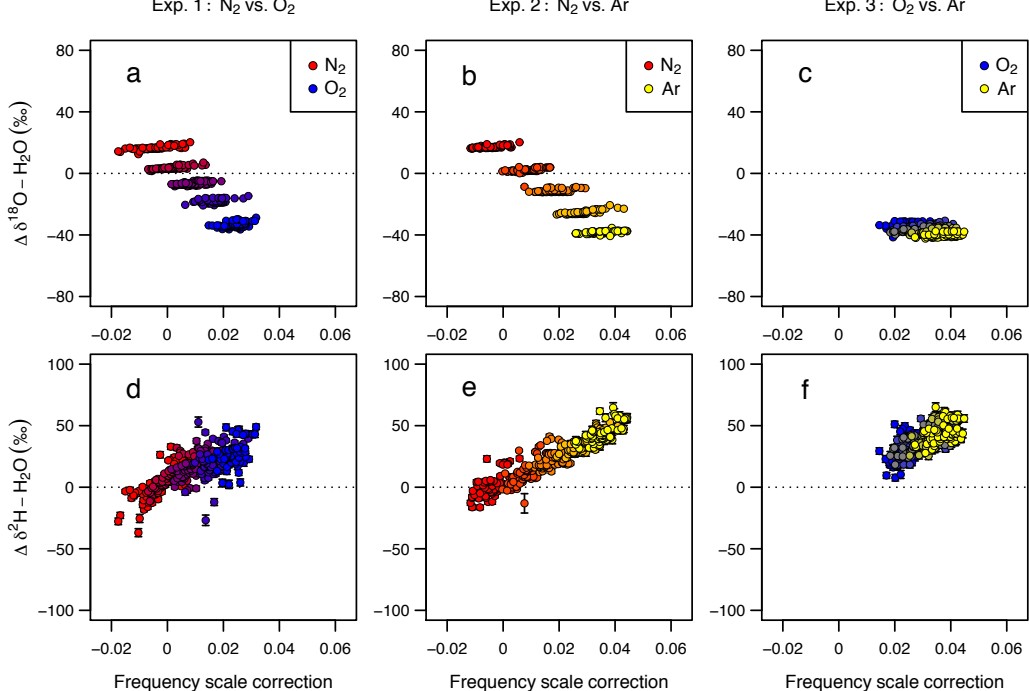


**Figure 8.** Relationship between the fixed linewidth parameter, $\Delta\delta^{18}O$-$H_2O$, and $\Delta\delta^2H$-$H_2O$. For each panel, n = 330 measurements; points represent mean values ± s.d.

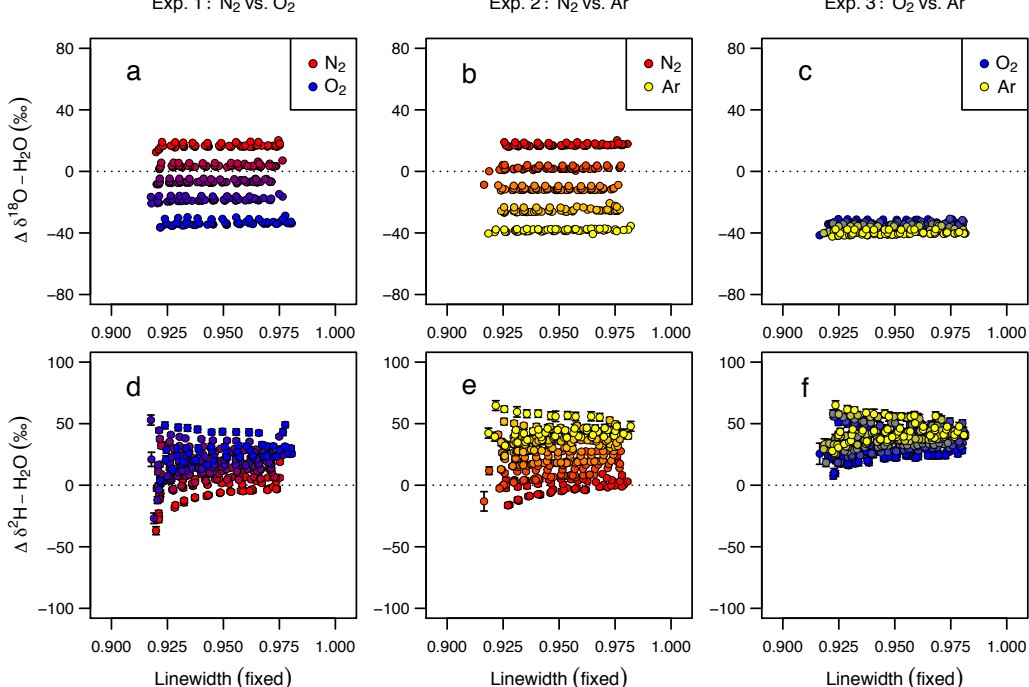



**Figure 9.** Relationship between the free linewidth parameter, $\Delta\delta^{18}$O-$H_2O$, and $\Delta\delta^2$H-$H_2O$. For each panel, n = 330 measurements; points represent mean values ± s.d.

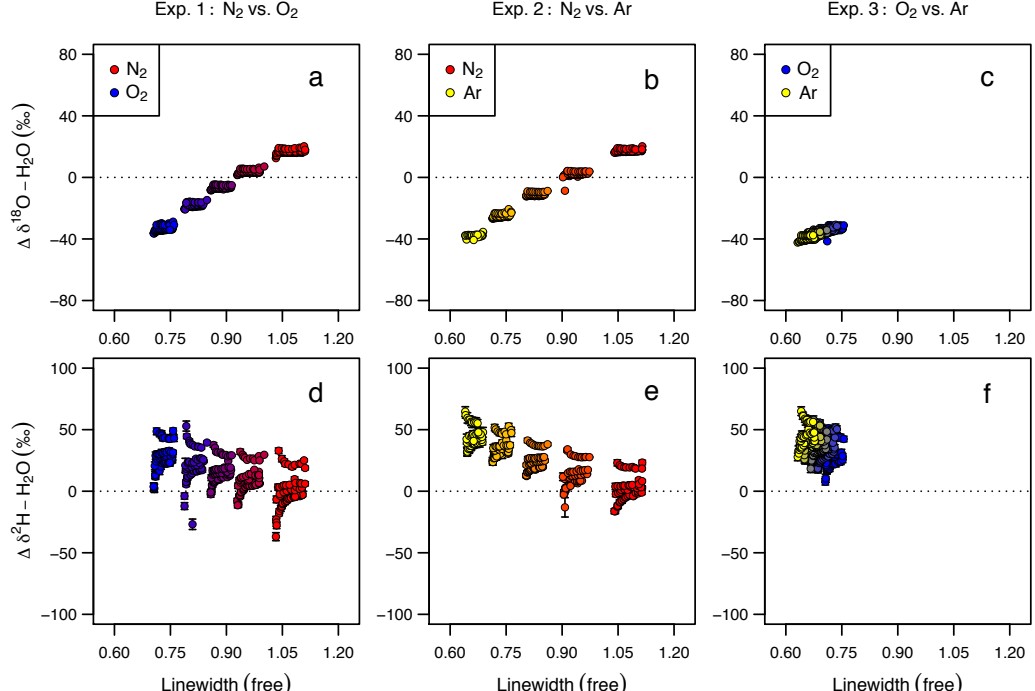




**Figure 10.** Relationships between $H_2O$ mixing ratio and the spectral residuals, baseline, and baseline slope

parameters. For each panel, n = 990 measurements; points represent mean values ± s.d.

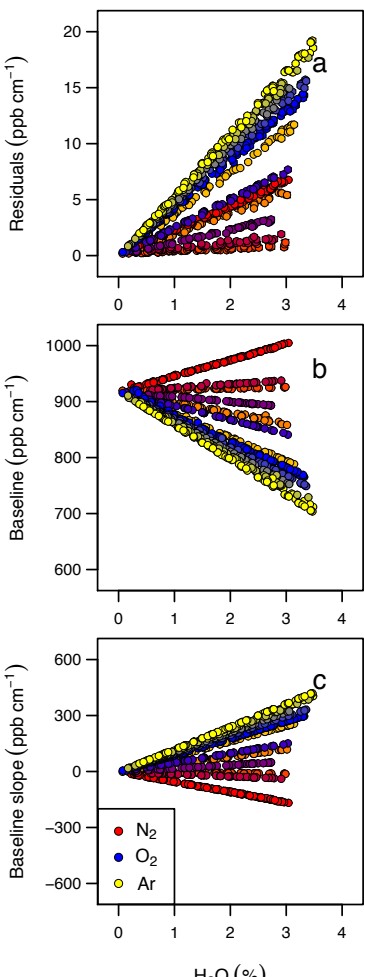



**Figure 11.** Relationships between $H_2O$ mixing ratio and the frequency shift, frequency scale correction, fixed linewidth, and free linewidth parameters. For each panel, n = 990 measurements; points represent mean values ± s.d.

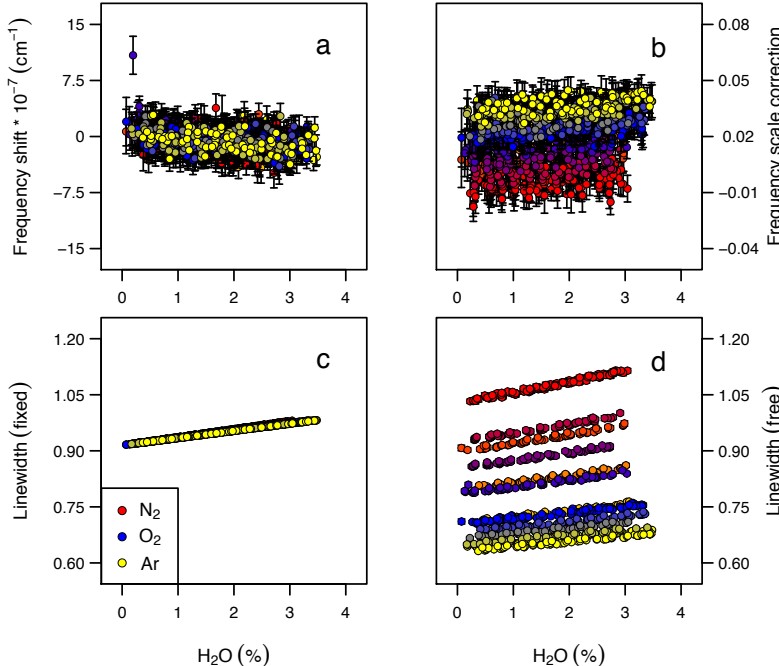



**Figure 12.** Direct and indirect effects of background gas composition on peak amplitude determination. Panels (a) and (b) plot theoretical spectrograms illustrating how isolated absorption features are directly broadened by air (solid line) versus $N_2$ (dashed line) or $O_2$ and Ar (dotted line). Panel (c) plots simulated spectrograms illustrating how the baseline of the three $H_2O$ lines targeted by the L2120-*i* is indirectly affected by the strong neighboring lines at lower frequencies. Simulations were performed using spectraplot.com with HITRAN/HITEMP data and the following parameters: 0.5 and 2.5 % $H_2O$, T = 80°C, P = 35 torr, and L = 1 cm.

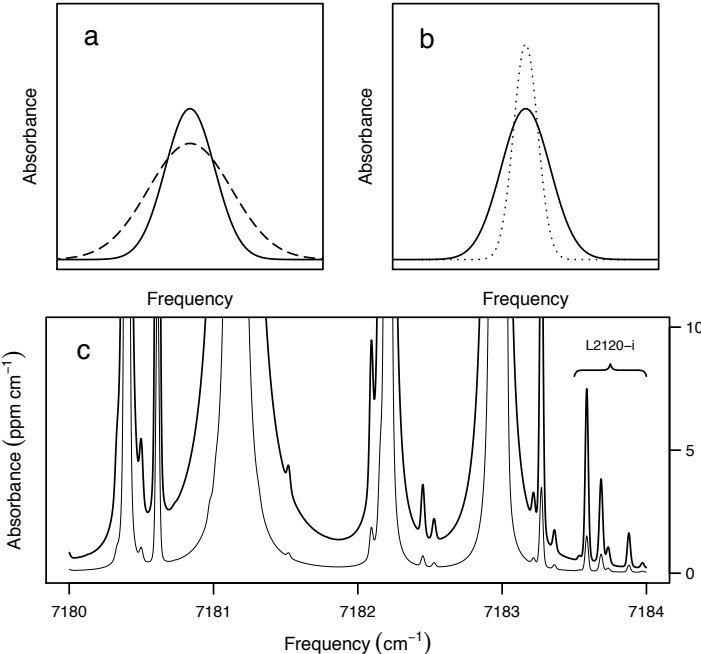





**List of Tables**

**Table 1.** Empirical models of the sensitivity of $\delta^{18}$O-$H_2O$ and $\delta^2$H-$H_2O$ values to $H_2O$ in pure $N_2$, $O_2$, and Ar. For significant predictors, coefficient estimates are given for $\beta_2$, $\beta_3$, and $\beta_4$ in Eq. 3 (see text). Overall model fit is summarized with the residual standard error (RSE), adjusted $R^2$ and P-value. Abbreviation: n.s., not significant.

**Table 2.** Empirical models of the sensitivity of $\delta^{18}$O-$H_2O$ and $\delta^2$H-$H_2O$ values to each of the binary mixtures of $N_2$, $O_2$, and Ar. For significant predictors, coefficient estimates are given for $\beta_5$ in Eq. 4 (see text). Overall model fit is summarized with the residual standard error (RSE), adjusted $R^2$ and P-value.

**Table 3.** Empirical models of the composite sensitivity of $\delta^{18}$O-$H_2O$ and $\delta^2$H-$H_2O$ values to all of the binary mixtures of $N_2$, $O_2$, and Ar. For significant predictors, coefficient estimates are given for $\beta_5$ and $\beta_6$ in Eq. 5 (see text). Overall model fit is summarized with the residual standard error (RSE), adjusted $R^2$ and P-value.

**Table 4.** Semi-mechanistic models of the composite sensitivity of $\delta^{18}$O-$H_2O$ and $\delta^2$H-$H_2O$ values to all of the binary mixtures of $N_2$, $O_2$, and Ar. For significant predictors, coefficient estimates are given for $\beta_7$ and $\beta_8$ in Eq. 6 (see text). Overall model fit is summarized with the residual standard error (RSE), adjusted $R^2$ and P-value.

**Table 5.** Comparison of empirical and semi-mechanistic models for predicting $\delta^{18}$O-$H_2O$ and $\delta^2$H-$H_2O$ values in a ternary mixture of $N_2$, $O_2$, and Ar. Overall model fit is summarized with the residual standard error (RSE), adjusted $R^2$ and P-value.




**Table 1.** Empirical models of the sensitivity of $\delta^{18}O$-$H_2O$ and $\delta^2H$-$H_2O$ values to $H_2O$ in pure $N_2$, $O_2$, and Ar. For significant predictors, coefficient estimates are given for $\beta_2$, $\beta_3$, and $\beta_4$ in Eq. 3 (see text). Overall model fit is summarized with the residual standard error (RSE), adjusted $R^2$ and P-value. Abbreviation: n.s., not significant.

| Model | Background | n | Response & Predictor | Mean ± SE (‰ ppm$^{-1}$) | RSE (‰) | Adj. $R^2$ | P-value |
|---|---|---|---|---|---|---|---|
| 1 | $N_2$ | 132 | $\partial(\delta^{18}O_{app})\,\partial(H_2O)^{-1}$ | -2.7e-04 ± 4.7e-05 | - | - | - |
| | | | $\partial(\delta^{18}O_{app})\,\partial(1/H_2O)^{-1}$ | -1.3e+04 ± 1.5e+03 | - | - | - |
| | | | $\partial(\delta^{18}O_{app})\,\partial(H_2O^2)^{-1}$ | 6.4e-09 ± 1.2e-09 | - | - | - |
| | | | - | - | 0.44 | 0.997 | < 0.001 |
| 2 | $N_2$ | 132 | $\partial(\delta^2H_{app})\,\partial(H_2O)^{-1}$ | n.s. | - | - | - |
| | | | $\partial(\delta^2H_{app})\,\partial(1/H_2O)^{-1}$ | -6.9e+04 ± 1.2e+04 | - | - | - |
| | | | $\partial(\delta^2H_{app})\,\partial(H_2O^2)^{-1}$ | 2.0e-08 ± 9.5e-09 | - | - | - |
| | | | - | - | 3.66 | 0.998 | < 0.001 |
| 3 | $O_2$ | 132 | $\partial(\delta^{18}O_{app})\,\partial(H_2O)^{-1}$ | -9.2e-05 ± 2.0e-05 | - | - | - |
| | | | $\partial(\delta^{18}O_{app})\,\partial(1/H_2O)^{-1}$ | -6.9e+03 ± 3.5e+02 | - | - | - |
| | | | $\partial(\delta^{18}O_{app})\,\partial(H_2O^2)^{-1}$ | 2.9e-09 ± 5.2e-10 | - | - | - |
| | | | - | - | 0.37 | 0.998 | < 0.001 |
| 4 | $O_2$ | 132 | $\partial(\delta^2H_{app})\,\partial(H_2O)^{-1}$ | 9.0e-04 ± 1.8e-04 | - | - | - |
| | | | $\partial(\delta^2H_{app})\,\partial(1/H_2O)^{-1}$ | n.s. | - | - | - |
| | | | $\partial(\delta^2H_{app})\,\partial(H_2O^2)^{-1}$ | -1.6e-08 ± 4.6e-09 | - | - | - |
| | | | - | - | 3.28 | 0.998 | < 0.001 |
| 5 | Ar | 132 | $\partial(\delta^{18}O_{app})\,\partial(H_2O)^{-1}$ | n.s. | - | - | - |
| | | | $\partial(\delta^{18}O_{app})\,\partial(1/H_2O)^{-1}$ | n.s. | - | - | - |
| | | | $\partial(\delta^{18}O_{app})\,\partial(H_2O^2)^{-1}$ | n.s. | - | - | - |
| | | | - | - | 1.16 | 0.983 | < 0.001 |
| 6 | Ar | 132 | $\partial(\delta^2H_{app})\,\partial(H_2O)^{-1}$ | n.s. | - | - | - |
| | | | $\partial(\delta^2H_{app})\,\partial(1/H_2O)^{-1}$ | n.s. | - | - | - |
| | | | $\partial(\delta^2H_{app})\,\partial(H_2O^2)^{-1}$ | n.s. | - | - | - |
| | | | - | - | 2.29 | 0.999 | < 0.001 |



**Table 2.** Empirical models of the sensitivity of $\delta^{18}O$-$H_2O$ and $\delta^2H$-$H_2O$ values to each of the binary mixtures of $N_2$, $O_2$, and Ar. For significant predictors, coefficient estimates are given for $\beta_5$ in Eq. 4 (see text). Overall model fit is summarized with the residual standard error (RSE), adjusted $R^2$ and P-value.

| Model | Background | n | Response & Predictor | Mean ± SE (‰ %$^{-1}$) | RSE (‰) | Adj. $R^2$ | P-value |
|---|---|---|---|---|---|---|---|
| 1 | $N_2$, $O_2$ | 330 | $\partial(\delta^{18}O_{app})\,\partial(O_2)^{-1}$ | -0.50 ± 0.001 | 0.62 | 0.999 | < 0.001 |
| 2 | $N_2$, $O_2$ | 330 | $\partial(\delta^2H_{app})\,\partial(O_2)^{-1}$ | 0.26 ± 0.009 | 5.19 | 0.995 | < 0.001 |
| 3 | $N_2$, Ar | 330 | $\partial(\delta^{18}O_{app})\,\partial(Ar)^{-1}$ | -0.56 ± 0.001 | 0.62 | 0.999 | < 0.001 |
| 4 | $N_2$, Ar | 330 | $\partial(\delta^2H_{app})\,\partial(Ar)^{-1}$ | 0.43 ± 0.005 | 3.16 | 0.998 | < 0.001 |
| 5 | $O_2$, Ar | 330 | $\partial(\delta^{18}O_{app})\,\partial(Ar)^{-1}$ | -0.06 ± 0.001 | 0.33 | 0.999 | < 0.001 |
| 6 | $O_2$, Ar | 330 | $\partial(\delta^2H_{app})\,\partial(Ar)^{-1}$ | 0.15 ± 0.005 | 3.33 | 0.998 | < 0.001 |





**Table 3.** Empirical models of the composite sensitivity of $\delta^{18}$O-$H_2O$ and $\delta^2$H-$H_2O$ values to all of the binary mixtures of $N_2$, $O_2$, and Ar. For significant predictors, coefficient estimates are given for $\beta_5$ and $\beta_6$ in Eq. 5 (see text). Overall model fit is summarized with the residual standard error (RSE), adjusted $R^2$ and P-value.

| Model | Background | n | Response & Predictor | Mean ± SE (‰ %$^{-1}$) | RSE (‰) | Adj. R$^2$ | P-value |
|-------|-----------|-----|----------------------|------------------------|---------|-----------|---------|
| 1 | $N_2$, $O_2$, Ar | 990 | $\partial(\delta^{18}O_{app})\,\partial(O_2)^{-1}$ | -0.50 ± 0.001 | - | - | - |
| | | | $\partial(\delta^{18}O_{app})\,\partial(Ar)^{-1}$ | -0.57 ± 0.001 | - | - | - |
| | | | - | - | 0.62 | 0.999 | < 0.001 |
| 2 | $N_2$, $O_2$, Ar | 990 | $\partial(\delta^2H_{app})\,\partial(O_2)^{-1}$ | 0.26 ± 0.004 | - | - | - |
| | | | $\partial(\delta^2H_{app})\,\partial(Ar)^{-1}$ | 0.42 ± 0.004 | - | - | - |
| | | | - | - | 3.59 | 0.998 | < 0.001 |



**Table 4.** Semi-mechanistic models of the composite sensitivity of $\delta^{18}O$-$H_2O$ and $\delta^2H$-$H_2O$ values to all of the binary mixtures of $N_2$, $O_2$, and Ar. For significant predictors, coefficient estimates are given for $\beta_7$ in Eq. 6 (see text). Overall model fit is summarized with the residual standard error (RSE), adjusted $R^2$ and P-value.

| Model | Background | n | Response & Predictor | Mean ± SE (‰ unit$^{-1}$) | RSE (‰) | Adj. $R^2$ | P-value |
|---|---|---|---|---|---|---|---|
| 1 | $N_2$, $O_2$, Ar | 990 | $\partial(\delta^{18}O_{app})\,\partial(\text{frequency shift})^{-1}$ | n.s. | 19.88 | 0.164 | < 0.001 |
| 2 | $N_2$, $O_2$, Ar | 990 | $\partial(\delta^{18}O_{app})\,\partial(\text{residuals})^{-1}$ | -5.0 ± 1.3e-01 | 12.80 | 0.653 | < 0.001 |
| 3 | $N_2$, $O_2$, Ar | 990 | $\partial(\delta^{18}O_{app})\,\partial(\text{fixed linewidth})^{-1}$ | 1.2e04 ± 2.0e02 | 9.20 | 0.821 | < 0.001 |
| 4 | $N_2$, $O_2$, Ar | 990 | $\partial(\delta^{18}O_{app})\,\partial(\text{baseline})^{-1}$ | 3.2e-01 ± 5.4e-03 | 9.17 | 0.822 | < 0.001 |
| 5 | $N_2$, $O_2$, Ar | 990 | $\partial(\delta^{18}O_{app})\,\partial(\text{baseline slope})^{-1}$ | -1.7e-01 ± 2.7e-03 | 9.11 | 0.824 | < 0.001 |
| 6 | $N_2$, $O_2$, Ar | 990 | $\partial(\delta^{18}O_{app})\,\partial(\text{frequency scale})^{-1}$ | -1.4e03 ± 1.6e01 | 6.72 | 0.904 | < 0.001 |
| 7 | $N_2$, $O_2$, Ar | 990 | $\partial(\delta^{18}O_{app})\,\partial(\text{free linewidth})^{-1}$ | 1.4e02 ± 5.8e-01 | 2.54 | 0.986 | < 0.001 |
| 8 | $N_2$, $O_2$, Ar | 990 | $\partial(\delta^2H_{app})\,\partial(\text{frequency shift})^{-1}$ | n.s. | 14.1 | 0.966 | < 0.001 |
| 9 | $N_2$, $O_2$, Ar | 990 | $\partial(\delta^2H_{app})\,\partial(\text{residuals})^{-1}$ | 3.4e00 ± 1.0e-01 | 9.7 | 0.984 | < 0.001 |
| 10 | $N_2$, $O_2$, Ar | 990 | $\partial(\delta^2H_{app})\,\partial(\text{baseline})^{-1}$ | -2.2e-01 ± 4.5e-03 | 7.7 | 0.990 | < 0.001 |
| 11 | $N_2$, $O_2$, Ar | 990 | $\partial(\delta^2H_{app})\,\partial(\text{baseline slope})^{-1}$ | 1.1e-01 ± 2.3e-03 | 7.7 | 0.990 | < 0.001 |
| 12 | $N_2$, $O_2$, Ar | 990 | $\partial(\delta^2H_{app})\,\partial(\text{fixed linewidth})^{-1}$ | -8.2e03 ± 1.6e02 | 7.4 | 0.990 | < 0.001 |
| 13 | $N_2$, $O_2$, Ar | 990 | $\partial(\delta^2H_{app})\,\partial(\text{frequency scale})^{-1}$ | 9.8e02 ± 1.1e01 | 4.9 | 0.996 | < 0.001 |
| 14 | $N_2$, $O_2$, Ar | 990 | $\partial(\delta^2H_{app})\,\partial(\text{free linewidth})^{-1}$ | -9.6e01 ± 1.0e00 | 4.4 | 0.997 | < 0.001 |





**Table 5.** Comparison of empirical and semi-mechanistic models for predicting $\delta^{18}$O-$H_2O$ and $\delta^2$H-$H_2O$ values in a ternary mixture of $N_2$, $O_2$, and Ar. Overall model fit is summarized with the residual standard error (RSE), adjusted $R^2$ and P-value.

| Model | Background | n | Response | Predictor | Model type | RSE (‰) | Adj. $R^2$ | P-value |
|---|---|---|---|---|---|---|---|---|
| 1 | $N_2$, $O_2$, Ar | 240 | $\delta^{18}O_{apparent}$ | $\delta^{18}O_{predicted}$ | Empirical | 0.99 | 0.988 | < 0.001 |
| 2 | $N_2$, $O_2$, Ar | 240 | $\delta^{18}O_{apparent}$ | $\delta^{18}O_{predicted}$ | Semi-mechanistic | 1.68 | 0.965 | < 0.001 |
| 3 | $N_2$, $O_2$, Ar | 240 | $\delta^2H_{apparent}$ | $\delta^2H_{predicted}$ | Empirical | 3.1 | 0.999 | < 0.001 |
| 4 | $N_2$, $O_2$, Ar | 240 | $\delta^2H_{apparent}$ | $\delta^2H_{predicted}$ | Semi-mechanistic | 3.0 | 0.999 | < 0.001 |
| 5 | $N_2$, $O_2$, Ar | 116 | $\delta^{18}O_{apparent}$ | $\delta^{18}O_{predicted}$ | Empirical | 0.80 | 0.992 | < 0.001 |
| 6 | $N_2$, $O_2$, Ar | 116 | $\delta^{18}O_{apparent}$ | $\delta^{18}O_{predicted}$ | Semi-mechanistic | 1.21 | 0.982 | < 0.001 |
| 7 | $N_2$, $O_2$, Ar | 116 | $\delta^2H_{apparent}$ | $\delta^2H_{predicted}$ | Empirical | 2.0 | 0.999 | < 0.001 |
| 8 | $N_2$, $O_2$, Ar | 116 | $\delta^2H_{apparent}$ | $\delta^2H_{predicted}$ | Semi-mechanistic | 2.1 | 0.999 | < 0.001 |