# Peer review of "Effects of variation in background mixing ratios of N2, O2, and Ar on the measurement of $\delta^{18}$ O-H2O and $\delta^{2}$ H-H2O values by cavity ring-down spectroscopy"

_Atmospheric Measurement Techniques, 2017_

## Referee Comment (RC1) · F. Aemisegger (Referee) · 19 May 2017

Review of "Effects of variation in background mixing ratios of $N_2$, $O_2$, and Ar on the measurement of $\delta^{18}O$-$H_2O$ and $\delta^2H$-$H_2O$ values by cavity ring-down spectroscopy" by Johnson and Rella 2017

This study presents a detailed analysis of the artifacts caused by variations in background mixing ratios for stable water isotope measurements using commercially available and largely used cavity ring-down spectrometers by Picarro (L2120-i). These artifacts are due to the influence of the background gas matrix on the exact shape of the target absorption lines. This paper very clearly shows that these background-induced effects can affect the measurement uncertainty when large variations in the carrier gas composition occur either during measurements and/or calibration. Possible correction schemes are discussed and the authors show that a correction to within the inherent measurement precision cannot be achieved with these schemes.

I recommend rapid publication of this very well written, important and scientifically relevant paper, which I found was a pleasure to read.

I have only a few minor comments that I listed in the following:

1) Check the order of citations and make sure they are cited chronologically (e.g. p. 2, L.8)
2) p. 2, L.32: Isn't it Demtröder?
3) p. 2, L. 35: "how specific CRDS analyzers are designed" do the authors mean in terms of hardware, software (i.e. scanning strategy)?
4) I do not fully understand why the authors always mention $\delta^{18}O$-$H_2O$ in pairs. Is this notation really justified and what does it exactly mean? For example on p. 4 at line 34 and in other instances.
5) p. 7, L. 30: What do "apparent" and "true" exactly mean? This was a bit confusing to me. Could this be defined more explicitly?
6) Eq. 3-5: Why do the author use this sort of linear multivariate model? Is there a physical justification for it?
7) p. 9, L. 25-30: Is this result likely to be consistent for different instruments of the same version?
8) Section 3.2 or elsewhere: could the authors add a short discussion about the impact of variations in the background gas composition on dexcess?
9) p. 12, L. 1-8: Could the authors compare the values in L. 1-2 to ambient values? Do these artifacts affect measurements in ambient near-surface and aircraft-based atmospheric water vapour? Or only in specific environments? This might have consequences for many current measurement applications and a short discussion on where to expect relevant impacts would be helpful.
10) p. 15, L. 7: What does "calibration-free" spectral fitting strategies mean? Could the authors add 1-2 sentences to explain this?
11) Very nice conclusions!
12) Fig. 2: It would be helpful to have a sentence in the caption saying what the different point for a given value on the x-axis mean. (I suppose $\Delta\delta^{18}O$ for different H2O mixing ratio levels?) Also here I am not sure whether the notation $\Delta\delta^{18}O$-$H_2O$ (‰) makes sense. As far as I understood it, I would opt for only $\Delta\delta^{18}O$-$H_2O$ (‰).
13) Fig. 12c: I did not understand the difference between the thick and the thin lines. Maybe the authors could mention it more clearly in the caption or in the panel.

---

## Author Comment (AC1) · 18 Jun 2017

Author response to Franziska Aemisegger's review of:

Johnson, J.E. and C.W. Rella, "Effects of variation in background mixing ratios of $N_2$, $O_2$, and Ar on the measurement of $\delta^{18}O$-$H_2O$ and $\delta^2H$-$H_2O$ values by cavity ring-down spectroscopy"

https://doi.org/10.5194/amt-2017-109

**Overall comment:** I recommend rapid publication of this very well written, important and scientifically relevant paper, which I found was a pleasure to read.

**Overall response:** Thank you for reviewing our manuscript. We appreciate the constructive comments, critiques, and questions. Below, we provide line-by-line responses to each of the specific points that you raised. We have incorporated the described revisions into a new version of the manuscript, and this is attached as a supplement.

**Comment 1**: Check the order of citations and make sure they are cited chronologically (e.g. p. 2, L.8)

**Response 1**: In the revised manuscript, the citations have been ordered chronologically.

**Comment 2**: p. 2, L.32: Isn't it Demtröder?

**Response 2**: Yes. Corrected.

**Comment 3**: p. 2, L. 35: "how specific CRDS analyzers are designed" do the authors mean in terms of hardware, software (i.e. scanning strategy)?

**Response 3**: Both. This sentence was meant as a transition into the next paragraph, which outlines the relevant hardware and software components. To clarify the meaning and improve the transition, we have revised the sentence in question to read: "However, whether or not errors actually occur in any given CRDS analyzer is a function of which specific absorption spectra are targeted, and how those spectra are acquired and interpreted."

**Comment 4**: I do not fully understand why the authors always mention $\delta^{18}O$-$H_2O$ in pairs. Is this notation really justified and what does it exactly mean? For example on p. 4 at line 34 and in other instances.

**Response 4**: On p. 4 at line 34, we reported the analytical precision of the standard measurements as "± 0.08 ‰ for $\delta^{18}O$-$H_2O$ and ± 0.9 ‰ for $\delta^2H$-$H_2O$." This notation means that the $\delta^{18}O$ and $\delta^2H$ values that are referred to are those of the $H_2O$ molecule, rather than of any other molecular species. This convention is useful when $\delta^{18}O$ and $\delta^2H$ values are being reported for multiple types of materials. However, since this manuscript deals only with the $\delta^{18}O$ and $\delta^2H$ values of water, the more complex notation is not strictly necessary. We have retained the references to $\delta^{18}O$-$H_2O$ and $\delta^2H$-$H_2O$ in the introduction where the L2120-$i$ is being introduced, but then in the remainder of the manuscript we have removed references to $\delta^{18}O$-$H_2O$ and $\delta^2H$-$H_2O$ and replaced them with the simpler terms $\delta^{18}O$ and $\delta^2H$.

**Comment 5**: p. 7, L. 30: What do "apparent" and "true" exactly mean? This was a bit confusing to me. Could this be defined more explicitly?

**Response 5**: To clarify the use of these terms, we have added definitions in the section 2.6.1. The new text reads: "To refer to calibrated $\delta^{18}O$ and $\delta^{2}H$ values as determined by IRMS, we use the subscript 'true' (i.e., $\delta^{18}O_{true}$ and $\delta^{2}H_{true}$). To refer to uncalibrated $\delta^{18}O$ and $\delta^{2}H$ values as determined by CRDS, we use the subscript 'apparent' (i.e., $\delta^{18}O_{apparent}$ and $\delta^{2}H_{apparent}$)." This notation is analogous to that used by Gralher et al. (2016) and facilitates comparison with the results of that study.

**Comment 6**: Eq. 3-5: Why do the author use this sort of linear multivariate model? Is there a physical justification for it?

**Response 6**: For Eq. 3, we simply selected the functional form based on previous work. Specifically, Rella had previously found that this expression provides a good description of the concentration-dependence of other Picarro analyzers that are similar to the L2120-*i*. We have added a sentence to clarify this: "This functional form provides a good description of the sensitivity of the apparent isotopic composition of water to the water mixing ratio in the L2120-*i* and similar analyzers from the same manufacturer (Rella 2010; Rella et al. 2015)." For Eq. 4 and 5, we then extended Eq. 3 with additional terms to capture the matrix effects. There was no *a priori* physical justification for this statistical approach; we used it only because it was the simplest way to summarize the sensitivity of each dependent variable (Y) to multiple independent variables (Xs).

**Comment 7**: p. 9, L. 25-30: Is this result likely to be consistent for different instruments of the same version?

**Response 7**: Previous work examining water vapor self-broadening within a constant background matrix has indicated that a water vapor correction function can be applied universally to a given model of WS-CRDS instrument as long as the calibrations are applied over the same scale (range) of water vapor mixing ratios (Chen et al. 2010 AMT 3: 374-386). Since self-broadening and matrix-broadening should affect all instruments in the same way, we expect that the interactions between self-broadening and matrix-broadening should also be consistent for different instruments of this same version. However, we are not aware of other published reports of the calibration coefficients that could be used to directly test this assumption. For example, the study of Gralher et al. (2016) is the closest point of comparison, but it focused on measurements made at a single $H_2O$ mixing ratio around 17,000 ppmv. As a result, those results do not provide a wide enough range of $H_2O$ values to reliably calculate how the coefficients of the $H_2O$ mixing ratio response vary between different inert gas backgrounds. We hope that publishing the calibration coefficients from our study will facilitate direct comparisons in future work.

**Comment 8**: Section 3.2 or elsewhere: could the authors add a short discussion about the impact of variations in the background gas composition on dexcess?

**Response 8**: Since the deuterium excess parameter is derived from $\delta^{18}O$ and $\delta^{2}H$ values, it will express the sensitivity of both the $\delta^{18}O$ and $\delta^{2}H$ values to the background gas composition. To emphasize this, we have revised

the text in the first paragraph of section 4.3, as follows: "[…] In combination, these findings indicate that there are several feasible approaches for *post hoc* calibrations of CRDS measurements that accurately account for background variation in $N_2$, $O_2$, and/or Ar, but that all currently tradeoff with measurement precision. Since the precision of the $\delta^{18}O$ and $\delta^2H$ measurements in turn controls the precision of the derived deterium excess parameter (i.e., *d*-excess =

5    $\delta^2H$ - 8*$\delta^{18}O$), this has important implications for the range of strategies that can be used to calibrate CRDS analyzers for $\delta^{18}O$, $\delta^2H$, and *d*-excess measurements. Different types of strategies are likely to be required for measurements: (i) in the atmosphere, and (ii) in other settings." The following two paragraphs then address the impact of variations in background gas composition on $\delta^{18}O$, $\delta^2H$, and *d*-excess, and calibration approaches that can be used to mitigate those impacts.

**Comment 9**: p. 12, L. 1-8: Could the authors compare the values in L. 1-2 to ambient values? Do these artifacts affect measurements in ambient near-surface and aircraft-based atmospheric water vapour? Or only in specific environments? This might have consequences for many current measurement applications and a short discussion on where to expect relevant impacts would be helpful.

15    **Response 9**: The range of natural variation in the $N_2$, $O_2$, and Ar content of the atmosphere is on the order of ~100 ppmv. Based on the L2120-i's sensitivities to variation in $N_2$, $O_2$, and Ar (p. 12, L. 1-2), this instrument is not expected to exhibit detectable isotope artifacts until the range of variation in $N_2$, $O_2$, and Ar is equal to or greater than ~5,000 ppmv. As a result, for atmospheric studies this issue is primarily important during calibration (and not during observations). In other settings where there is greater natural variation in in $N_2$, $O_2$, and Ar, this issue may be

20    important both during calibration and during observations. To make this point more clearly and quantitatively, we have revised the text in the Introduction as follows: "To date, background effects on CRDS measurements have been reported in three different types of situations. First, calibrations for observations of the unconfined atmosphere: even though the natural levels of variability in atmospheric $N_2$, $O_2$, and Ar mixing ratios are small (i.e., ~100 ppmv), large contrasts can occur between the average composition of the atmosphere and the composition of the mixtures used

25    for calibration (i.e., ~10,000 ppmv; Chen et al., 2010; Aemisegger et al., 2012; Nara et al., 2012; Long et al., 2013). Second, observations of confined atmospheres: for trace gas measurements in lakes, streams, oceans, and soils, the background concentrations of $O_2$ can vary naturally over an even wider range because the rates of biological processes that produce and consume this gas can proceed more rapidly than the physical processes that control mixing with the unconfined atmosphere (i.e., ~150,000 ppmv; Friedrichs et al., 2010; Becker et al., 2012)."

**Comment 10**: p. 15, L. 7: What does "calibration-free" spectral fitting strategies mean? Could the authors add 1-2 sentences to explain this?

**Response 10**: The term "calibration-free" generally refers to a spectral fitting strategy that does not rely on an *in situ* signal calibration. Here, we are using the term to highlight a specific approach in which all of the lineshape

35    parameters are treated as free variables during spectral fitting, rather than being fixed according to an *a priori* calibration. To clarify, we have revised this paragraph as follows: "[…] while measurements of integrated absorbance are likely to be necessary for limiting sensitivity to background effects, they are unlikely to be sufficient

for entirely eliminating sensitivity to background effects. To achieve this objective, it may be useful to introduce spectral fitting strategies in which all of the lineshape parameters are treated as free rather than fixed variables. Such 'calibration-free' spectral fitting strategies have been recently been developed for high-temperature and high-pressure applications in energy research, and these might serve as models for lower-temperature and lower-pressure applications in environmental research (Sun et al., 2013; Goldenstein et al., 2014; Sur et al., 2015; Goldenstein et al., 2017)."

**Comment 11**: Very nice conclusions!
**Response 11**: Thank you.

**Comment 12, part 1**: Fig. 2: It would be helpful to have a sentence in the caption saying what the different point for a given value on the x-axis mean. (I suppose $\Delta\delta^{18}O$ for different H2O mixing ratio levels?)
**Response 12, part 1**: Yes, exactly. For a given value on the x-axis, the different points represent measurements of the four liquid standards across a range of injection volumes. We have revised the caption of Fig. 2 to read: "For each panel, n = 330 measurements of four liquid standards across a range of injection volumes (i.e., 400-2400 nL, in eleven steps of 200 nL each). Points represent mean values ± s.d. for replicates of each standard at each injection volume." We have also revised the other figure captions in an analogous way.

**Comment 12, part 2**: Also here I am not sure whether the notation $\Delta\delta^{18}O$-$H_2O$ (‰) makes sense. As far as I understood it, I would opt for only $\Delta\delta^{18}O$-$H_2O$ (‰).
**Response 12, part 2**: This comment is a little difficult to understand because the two notations appear to be identical within the referee comment – perhaps there is a typo? Based on the referee's comment 4, we have replaced the "$\Delta\delta^{18}O$-$H_2O$" notation with the simplified notation "$\Delta\delta^{18}O$." Hopefully this revision address the original concern satisfactorily. However, if we have misunderstood the original concern, please advise us.

**Comment 13**: Fig. 12c: I did not understand the difference between the thick and the thin lines. Maybe the authors could mention it more clearly in the caption or in the panel.

[revised manuscript text omitted]

---

## Referee Comment (RC2) · Anonymous Referee #2 · 30 Jun 2017

This paper describes an interesting test of the ability of a particular make of cavity-ring down spectrometer (the L2120-i by Picarro, Inc.) to accurately assess the concentration of isotopic species of water in the presence of backgrounds of molecular nitrogen and oxygen and of atomic argon. The authors describe a series of carefully controlled measurements that reveal subtle deviations of the true isotopic content from that reported by the spectrometer as a consequence of changing the concentration and composition of background gases. An empirical model is proposed and used to assess the sensitivity of the deviations to various powers of the water mixing ratio, as well as to

mixing ratio of the background gases. The model is believable and reveals a great deal about the trends of the deviations.

Papers such as these are important both to provide a critical assessment of the capability of an instrument to make research-grade measurements, and to establish a methodology for the assessment. Since the paper is very clearly written and logically organized, it achieves these objectives. Through multiple readings I have found nothing in either the text or in the figures that requires change. I expect that the community will find this document useful, and recommend that it be published in its current form.

---

## Author Comment (AC2) · 30 Jun 2017

Thank you for your review of this manuscript. Given that you did not recommend any revisions in the text or figures, we have not made any further adjustments to the manuscript.